# The cyanobacterial circadian clock follows midday in vivo and in vitro

Eugene Leypunskiy[1], Jenny Lin[2], Haneul Yoo[2], UnJin Lee[3], Aaron R Dinner[1,4,5]*, Michael J Rust[1,3,6,7]*

[1]Graduate Program in Biophysical Sciences, The University of Chicago, Chicago, United States; [2]Department of Biochemistry and Molecular Biology, The University of Chicago, Chicago, United States; [3]Department of Ecology and Evolution, The University of Chicago, Chicago, United States; [4]Department of Chemistry, The University of Chicago, Chicago, United States; [5]James Franck Institute, The University of Chicago, Chicago, United States; [6]Department of Molecular Genetics and Cell Biology, The University of Chicago, Chicago, United States; [7]Department of Physics, The University of Chicago, Chicago, United States

**Abstract** Circadian rhythms are biological oscillations that schedule daily changes in physiology. Outside the laboratory, circadian clocks do not generally free-run but are driven by daily cues whose timing varies with the seasons. The principles that determine how circadian clocks align to these external cycles are not well understood. Here, we report experimental platforms for driving the cyanobacterial circadian clock both in vivo and in vitro. We find that the phase of the circadian rhythm follows a simple scaling law in light-dark cycles, tracking midday across conditions with variable day length. The core biochemical oscillator comprised of the Kai proteins behaves similarly when driven by metabolic pulses in vitro, indicating that such dynamics are intrinsic to these proteins. We develop a general mathematical framework based on instantaneous transformation of the clock cycle by external cues, which successfully predicts clock behavior under many cycling environments.

*For correspondence: dinner@uchicago.edu (ARD); mrust@uchicago.edu (MJR)

**Competing interests:** The authors declare that no competing interests exist.

## Introduction

Circadian clocks generate biological rhythms that temporally organize physiology to match the 24 hr diurnal cycle. Although these clocks continue to oscillate in constant laboratory conditions, in natural environments they are driven by the external cycle of day and night. Because of latitude-dependent changes in the duration of day and night throughout the year, clock architectures must incorporate mechanisms that respond appropriately to environmental signals, such as dawn and dusk, whose schedule changes throughout the year. Data from several species indicate that circadian clocks adapt to different day lengths by modulating their phases relative to the light-dark cycle (*de Montaigu et al., 2015*; *Rémi et al., 2010*). In this way, circadian oscillators are able to coordinate physiological events relative to specific times of day and, in multicellular organisms, specialized mechanisms exist that allow oscillators in different cells to follow distinct times of day (*Herzog, 2007*; *Daan et al., 2003*). Because the biochemical circuits governing circadian rhythms and light-dark sensing in such organisms are complex, it has been difficult to identify which features of circadian systems are responsible for seasonal adaptation. More generally, it remains unclear what features oscillators must have to respond appropriately to varying light-dark cycles and how those features are implemented molecularly in specific systems.

Cyanobacteria present a unique opportunity to elucidate molecular mechanisms in circadian biology because the core oscillator responsible for driving genome-wide transcriptional rhythms can be

**eLife digest** All life forms on the surface of our planet face an environment that cycles between day and night as the Earth rotates around its axis. To deal with these regular changes, organisms from bacteria to humans have internal rhythms, called circadian clocks, that coordinate different aspects of these organisms' lives, from growth to the urge to sleep, with the day-night cycle.

The simplest of all known circadian clocks is found in bacteria called cyanobacteria, which live in bodies of water all around the world. Like plants, these microorganisms can harness the energy in sunlight to fuel their growth. The internal clock of cyanobacteria is remarkable because it can be rebuilt in the laboratory using just three components, proteins called KaiA, KaiB and KaiC. When mixed together in a test tube, these proteins spontaneously generate a 24-hour cycle in which KaiC gets chemically modified by the addition and removal of a phosphate group. These chemical modifications on KaiC are like the hands of the clock that tell the time of day.

Much work on the internal clock of cyanobacteria has focused on how the clock proteins generate rhythms. What has been less clear is how this rhythm works in an environment that changes between day and night and from season to season. For example, away from the equator, the length of the day changes throughout the year. Changes in day length are important for cyanobacteria because they can grow only during the day and must rest at night. Yet, it was not understood how this microorganism's clock adjusts to days of different length.

To address this question, Leypunskiy et al. made a device to grow cyanobacteria in many different light-dark cycles while monitoring their circadian rhythm. The experiments revealed a fairly simple result: the internal clock time is always the same at the middle of the day, regardless of how long the day is. Next, Leypunskiy et al. developed a way to study how the clock proteins in the test tube respond to chemical signals mimicking day and night. Unexpectedly, this showed that the Kai proteins themselves have the ability to track the middle of the day. It turns out that these proteins are not only able to generate a 24-hour rhythm, but also to correctly align it to the day-night cycle.

Together, these findings will guide other researchers to better understand the molecular origins of how circadian clocks align to the day-night cycle. Moreover, by showing that the effects of a light-dark cycle on a circadian clock can be mimicked in a test tube, these results open the doors to using the tools of biochemistry to understand how circadian clocks work in natural environments.

reconstituted using purified KaiABC proteins, and these proteins have been extensively studied biochemically in constant conditions (*Nishiwaki et al., 2004*; *Rust et al., 2007*). Cyanobacteria must contend with large seasonal variations in day length because their natural aquatic environments span a wide range of latitudes (*Flombaum et al., 2013*). Yet, comparatively little is known about how the cyanobacterial clock functions when driven by light-dark cycles that mimic days in different seasons.

To study how the circadian clock is affected by light-dark cycles with different day lengths, we developed multiplexed LED illumination devices to grow cyanobacteria in a wide range of light-dark conditions (*Figure 1A*). We used square-wave illumination patterns in our experiments, and used the times of lights-on and lights-off as experimental analogs of dawn and dusk, respectively. We defined the day length ($\tau$) as the total time the lights are on each day (*Figure 1A*). We found that the circadian rhythm in the cyanobacterium *Synechococcus elongatus* PCC 7942 (*S. elongatus*) follows a simple rule: the phase of clock-driven gene expression scales linearly with day length and remains fixed relative to the middle of the day over a wide range of day lengths.

We biochemically simulated seasonal effects in the reconstituted KaiABC oscillator by driving the Kai proteins with pulses of nucleotides on a 24 hr cycle, simulating in vivo metabolic conditions during day and night. Remarkably, this in vitro system shows a similar linear scaling of clock phase with simulated day length, indicating that core mechanisms for seasonal adaptation are intrinsic to the Kai proteins. We developed a minimal mathematical model for oscillator entrainment based on rapid transitions between distinct day and night clock cycles. The model can account for the ability to track midday if the phase shifts caused by dawn and dusk depend linearly on clock time with appropriate slopes. By calibrating this model with independent measurements in vitro and in vivo, we were able

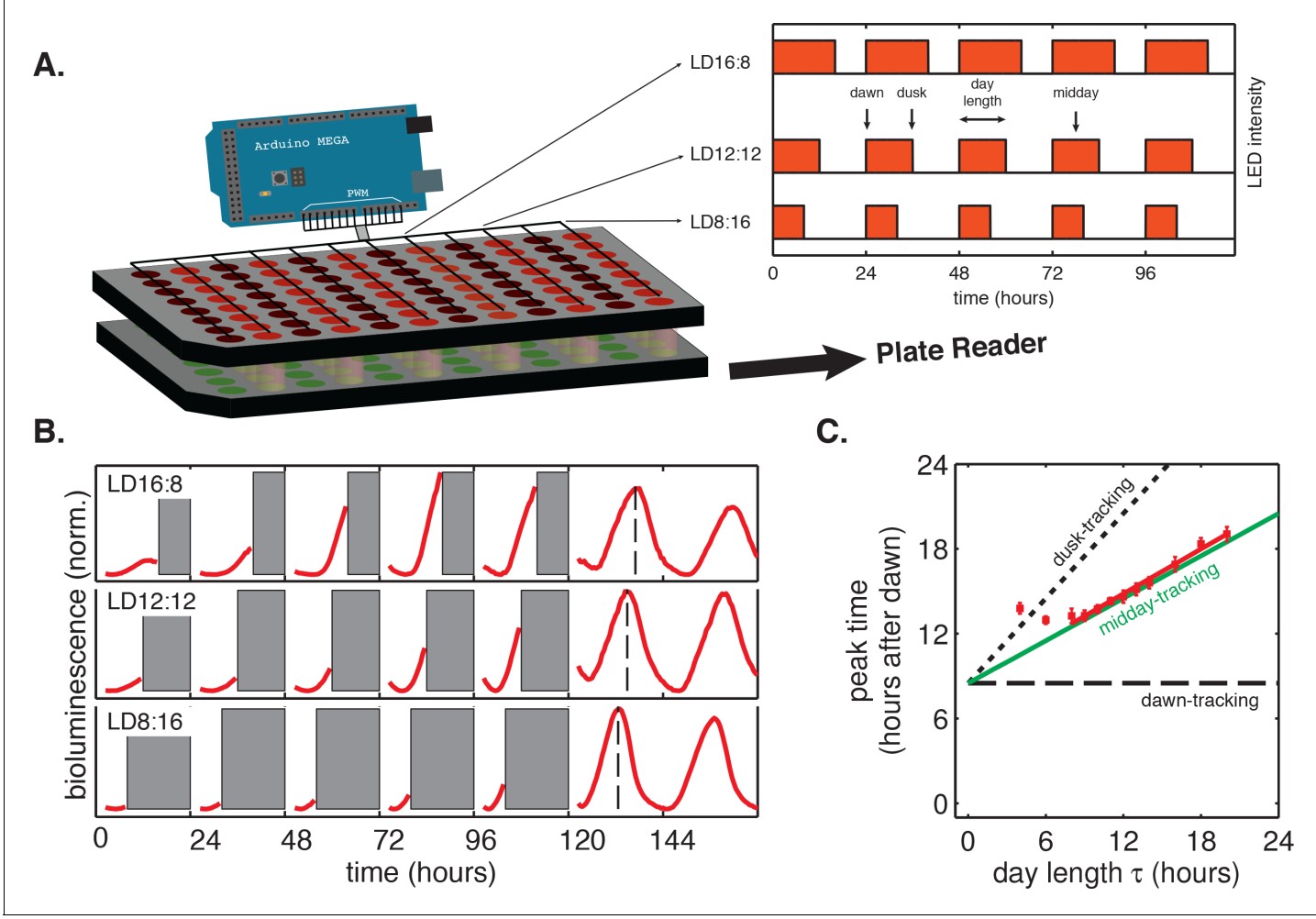

**Figure 1.** Phase of the cyanobacterial circadian rhythm scales linearly with day length. (**A**) LED array device used to grow *S. elongatus* in programmable light-dark cycles. Cells grown in a 96-well plate on solid media (lower plate, *green circles*) are illuminated from above by LEDs (*red circles*). An Arduino microcontroller is used to dynamically change LED intensity in different columns of the plate (*inset*). Luminescence from the bottom plate is read out every 30 min on a plate reader. Drawing not to scale. (**B**) Drive-and-release strategy to measure phase of the circadian clock under light-dark (LD) cycling. Cells were exposed to five entraining LD cycles and then released into constant light. Bioluminescence signals ($P_{kaiBC}$::*luxAB*) from each well were separated into individual 'day' and 'night' windows. Data from night portions of the experiment were omitted from analysis (*gray bars*), and data from the day portions of the experiment were aligned to zero baseline and normalized to unit variance. Dashed lines indicate time of peak reporter signal calculated by parabolic fitting. See Computational methods for details. (**C**) Peak time of bioluminescence ($P_{kaiBC}$::*luxAB*) in light-dark cycles of different day length (*red squares*) was quantified by local parabolic fitting around the first maximum of the oscillation after release into constant light. Error bars represent standard deviations of peak time estimates from technical replicates (n = 4–8). Slope of the linear fit (*red line*, m = 0.53 ± 0.01) was determined by linear regression. *Dashed and dotted black lines* indicate scaling of phase with day length for dawn- and dusk-tracking oscillators; *green line* indicates midday-tracking behavior.

The following source data and figure supplements are available for figure 1:

**Source data 1.** Source data for *Figure 1B*.
**Source data 2.** Source data for *Figure 1C*.
**Source data 3.** Source data for bioluminescence trajectories in *Figure 1—figure supplement 1*.
**Source data 4.** Source data for Kendall's τ correlations in *Figure 1—figure supplement 1*.
**Source data 5.** Source data for *Figure 1—figure supplement 2A–E*.

*Figure 1 continued*

**Source data 6.** Source data for *Figure 1—figure supplement 3A*, showing bioluminescence output from the *purF* repoter.

**Source data 7.** Source data for *Figure 1—figure supplement 3B*.

**Figure supplement 1.** Bioluminescence recordings from $P_{kaiBC}$::*luxAB* reporter in light-dark cycles.

**Figure supplement 2.** The circadian rhythm of *S. elongatus* rapidly entrains to 24 hr diurnal cycles with 8–16 hr of daylight.

**Figure supplement 3.** Bioluminescence recordings from $P_{purF}$::*luxAB* reporter in light-dark cycles.

to predict the results of phase response experiments, and the behavior of the clock in non-24 hr light-dark cycles.

The essential feature of the model, the linear dependences of phase shifts on clock time, has a simple geometric interpretation in terms of the deformation of the clock orbit caused by a transition between light and dark. Thus, the entrained behavior of the circadian clock can be captured quantitatively by a mathematical framework with a small number of parameters. This framework in turn can be used to guide the design of experiments that probe the molecular origins of key mathematical features. The precisely defined nature of the cyanobacterial clock facilitates such experiments, but the model is general, so we expect it to be useful for other organisms as well.

## Results

### Cyanobacteria respond to seasonal changes in day length by aligning the phase of circadian gene expression relative to midday

*S. elongatus* is a photosynthetic bacterium whose physiology is closely tied to light and dark. In constant light, the circadian clock exerts pervasive control over gene expression: most transcripts cycle in abundance with ≈ 24 hr period, with the majority of transcripts peaking either near subjective morning (dawn genes), or nearly 12 hr later, at subjective nightfall (dusk genes) (*Liu et al., 1995*; *Ito et al., 2009*). In the dark, however, growth stops, and most gene expression is highly repressed (*Hosokawa et al., 2011*; *Tomita et al., 2005*; *Pattanayak et al., 2015*). Thus, portions of the circadian gene expression program that fall into nighttime hours are strongly attenuated.

Viewed in this way, the hours between dawn and dusk provide a limited window for gene expression, so that winter months at high latitude provide fewer daylight hours for this cyanobacterium to accomplish biosynthetic tasks relative to summer months. We surmised that an important function for the circadian clock in this organism might be to schedule gene expression appropriately during daylight hours, when biosynthetic resources are available. We thus expected that asymmetric light-dark cycles mimicking days in different seasons would realign the clock cycle.

To systematically study how circadian gene expression in *S. elongatus* adjusts to asymmetric light-dark schedules, we built a microcontroller-driven LED array device and used it to grow cells in 24 hr day-night cycles with photoperiod (day length) varying from 4 to 20 hr (*Figure 1A*). This LED array was coupled to a plate reader that allowed us to monitor clock output—gene expression from clock-driven promoters—using strains engineered with luminescent reporters of representative dusk (*kaiBC*) and dawn (*purF*) genes.

In these experiments, the circadian clock stably entrained to light-dark cycles with a wide range of day lengths (τ = 8–16 hr) within roughly 3 days (*Figure 1—figure supplements 1* and *2*). We found that the phase of the oscillation, that is, when the peak clock output signal occurred relative to dawn, varied systematically as a function of day length. In general, longer days resulted in the peak reporter signal occurring later in the day (*Figure 1B*). By plotting the time of peak reporter signal versus day length, we found that the driven behavior of the clock follows a simple scaling law: the clock phase is proportional to day length (*Figure 1C*). The proportionality constant, *m*, falls in an intermediate range (*Table 1*) between the limits corresponding to either dawn (*m* = 0) or dusk (*m* = 1) fully resetting the clock. The linear relationship between clock phase and day length with slope *m* ≈ 0.5 implies that cyanobacteria set their clocks to reach the same internal time at the

**Table 1.** Summary of biologically independent in vivo experiments measuring entrainment to 24 hr light-dark cycles of varying day length and corresponding estimates of $m$, the proportionality coefficient between the peak time of $P_{kaiBC}$::luxAB reporter and day length during light-dark entrainment.

| Figure | Driving period $T$ (hr) | Day length $\tau$ (hr) | Slope $m \pm$ SD of estimate |
|---|---|---|---|
| *Figure 1C* | 24 | 4, 6, 8, 9, 10, 11, 12, 13, 14, 16, 18, 20 | 0.55 ± 0.02 (sinusoidal fitting)<br>0.53 ± 0.01 (parabolic fitting) |
| *Figure 1—figure supplement 2* | 24 | 8, 12, 16 | 0.47 ± 0.03 (sinusoidal fitting)<br>0.57 ± 0.02 (parabolic fitting) |
| *Figure 5C* | 22, 23, 24, 25, 26 | 8, 10, 12, 14 | 0.51 ± 0.11 (sinusoidal fitting) |

middle of the daylight hours, independent of day length. This entrainment behavior is not unique to the *kaiBC* promoter; a reporter for *purF*, a representative of the dawn class of genes, shows similar scaling behavior (*Figure 1—figure supplement 3*).

The fact that the core oscillator of the cyanobacterial circadian clock can be reconstituted in vitro from purified Kai proteins (*Nakajima et al., 2005*; *Kageyama et al., 2006*) naturally led to the question of whether the ability to track midday in vivo is intrinsic to the core oscillator, or if it requires additional factors present in the cell. We thus sought to extend the reconstituted system such that we could drive the biochemical oscillator with rhythmic input signals.

## In vitro reconstitution of seasonal clock response

A purified mixture of the KaiA, KaiB, and KaiC proteins spontaneously generates a stable circadian rhythm in KaiC phosphorylation and in the formation of KaiB-KaiC complexes (*Nakajima et al., 2005*; *Kageyama et al., 2006*). Although the Kai proteins are not light-sensitive, recent work has shown that they are sensitive to metabolite pools that shift in response to changes in photosynthetic activity in the cell: KaiA is sensitive to the redox state of quinones, and KaiC phosphorylation is sensitive to the ATP/ADP ratio (*Rust et al., 2011*; *Kim et al., 2012*; *Phong et al., 2013*). We thus developed a protocol to mimic repeated light-dark cycles in vitro by cycling the ATP/ADP ratio between physiologically relevant levels experienced during the day and night (*Pattanayak et al., 2014*), using ADP addition to simulate nightfall ([ATP]/([ATP]+[ADP]) $\approx$ 25%) followed by buffer exchange into an ATP-only buffer to simulate dawn ([ATP]/([ATP]+[ADP]) $\approx$ 100%) (*Figure 2A*).

We found that the KaiC phosphorylation rhythm readily responded to this metabolite pulsing protocol (*Figure 2B*). Stepping down to lower ATP/ADP conditions promoted dephosphorylation, in accord with previously published observations (*Pattanayak et al., 2014*). Conversely, stepping up to higher ATP/ADP conditions favored phosphorylation (*Figure 2B*). Pulsing the ATP/ADP ratio in this way caused the KaiC phosphorylation rhythm to synchronize to the external cycle. By the third cycle, the oscillator appeared to have stably entrained, and, similar to the clock behavior that we observed in live cells, longer times in daytime buffer led to later peak phosphorylation times. We estimated peak times for these oscillator reactions during the third entraining cycle and found that the phase of the in vitro rhythm scales linearly in proportion with simulated day length (*Figure 2C*), with KaiC phosphorylation rhythm peaking roughly 3 hr after the midpoint of simulated daytime, suggesting that the reconstituted Kai oscillator is capable of tracking the approximate midday point of an externally imposed metabolic rhythm.

To more accurately measure the scaling of entrained phase with day length in KaiABC oscillator reactions, we turned to a fluorescence polarization probe that enables automated measurement of oscillator state with high temporal resolution (*Figure 2—figure supplement 1*) (*Chang et al., 2012*; *Heisler et al., 2017*). We used this assay to determine clock phases in free-running conditions after entraining the Kai proteins with three metabolic cycles, analogous to the design of the in vivo experiments. We again measured linear scaling of the entrained phase, albeit with a lower slope than the estimate from sparser gel-based measurements of KaiC phosphorylation ($m = 0.38 \pm 0.07$ for polarization probe vs. $m = 0.51 \pm 0.04$ for phosphorylation data, *Figure 2C*). Despite the variability in our estimates of $m$, these measurements suggest that the in vitro oscillator successfully captures the essential feature of seasonal entrainment we observed in vivo: linear scaling of entrained phase with an intermediate slope.

In the cell, the Kai proteins interact with histidine kinases and a network of other factors absent from the reconstituted system, ultimately leading to rhythms in transcription across the genome, including the *kai* genes themselves (*Takai et al., 2006*; *Gutu et al., 2013*; *Markson et al., 2013*). These additional factors may account for the differences we observe between proportionality constants in vitro and in vivo, and the time delay between KaiC phosphorylation and luminescent output of the *kaiBC* expression reporter is likely responsible for the offset in *Figure 2C*. While these considerations suggest that care must be taken in connecting the in vitro and in vivo results, they also underscore the significance of our observation that the Kai oscillator proteins by themselves are sufficient to yield the linear response of the system to altered day length. We therefore sought to use this in vitro model of seasonality to uncover the biochemical basis of the linear seasonal clock response.

## Seasonal adaptation of the circadian oscillator can be decomposed into step responses to individual metabolic cues

In the purified system, the entraining cues are steps between high and low ATP/ADP that simulate dawn and dusk. We therefore sought to decompose the seasonal adaptation of the clock into step responses following each metabolic transition. This approach is related to the limit cycle theory of oscillators: if we assume that a unique stable orbit exists for both the day and night conditions, then a metabolic transition forces the system to adjust from its current cycle to one associated with the new condition. If relaxation to the new orbit occurs rapidly relative to the length of the day—that is, if the light and dark orbits are strongly attracting—then the state of the oscillator at each transition can be specified simply by a phase angle on that orbit. In this limit, the response of the system is fully determined by instantaneous phase shifts caused by the transitions. Mathematically, we describe this limit as an oscillator comprised of a single phase variable, $\theta$, that runs along a fixed limit cycle trajectory at constant speed (see Appendix 1). The responses to dark-to-light or light-to-dark transitions are then represented by phase shifts that are specified by the functions $L(\theta)$ and $D(\theta)$, respectively.

To determine whether this fast-relaxation approximation can describe phase shifts in the core Kai oscillator, and to study their biochemical basis, we measured the $L(\theta)$ and $D(\theta)$ shift functions in our reconstituted system. To measure the $D(\theta)$ function, we incubated the Kai proteins in daytime buffer, then transferred them into nighttime buffer at various times throughout the circadian cycle and studied how the oscillation was affected. As above, we assayed oscillator response in separate experiments using either SDS-PAGE to measure KaiC phosphorylation or the fluorescence polarization probe to achieve high temporal resolution. Example time series from the fluorescence polarization measurements are shown in *Figure 3A–B*.

We proceeded to calculate phase shifts from these measurements, taking into consideration that a switch to nighttime buffer conditions could affect the oscillator in two ways: the phase of the oscillation can shift, and the oscillator period may change, both of which can change the relative timing of peaks. To separate these effects, it is necessary to collect enough data to accurately determine the period, and then extrapolate backwards to infer the instantaneous phase shift at the time of buffer exchange. We detail our procedure for analyzing these data to extract phase shifts in *Figure 3—figure supplement 1*. We used an analogous design to measure the reverse steps from nighttime to daytime buffer (*Figure 3B*). Plotted together in *Figure 3C–D*, the extrapolated phase shifts measure the sensitivity of the clock to individual light and dark steps throughout the 24 hr day.

The measured $L(\theta)$ and $D(\theta)$ step-response functions show that the phase of the purified KaiABC oscillator is responsive to both step-up and step-down metabolic changes. For example, a day-to-night shift lengthens the period of the clock by about 2 hr, and, when it is applied at subjective morning, leads to an instantaneous phase delay of about 3 hr (*Figure 3—figure supplement 1*). On the other hand, a night-to-day transition in the middle of subjective night shortens the period and causes a phase advance of about 1.5 hr.

These measurements characterize how the oscillator responds to a metabolic step-change throughout the entire clock cycle. However, when the clock is stably synchronized to a cycling environment, dawn and dusk fall only in a limited window of clock phases. To determine this window, we returned to our measurements in *Figure 2C*. We identified phases that correspond to dawn or dusk in entrained conditions mimicking LD 6:18 to LD 18:6 and marked these phases on the $L(\theta)$ and $D(\theta)$ functions (*Figure 3C–D*, colored arrows). These points fell on gradually changing, approximately

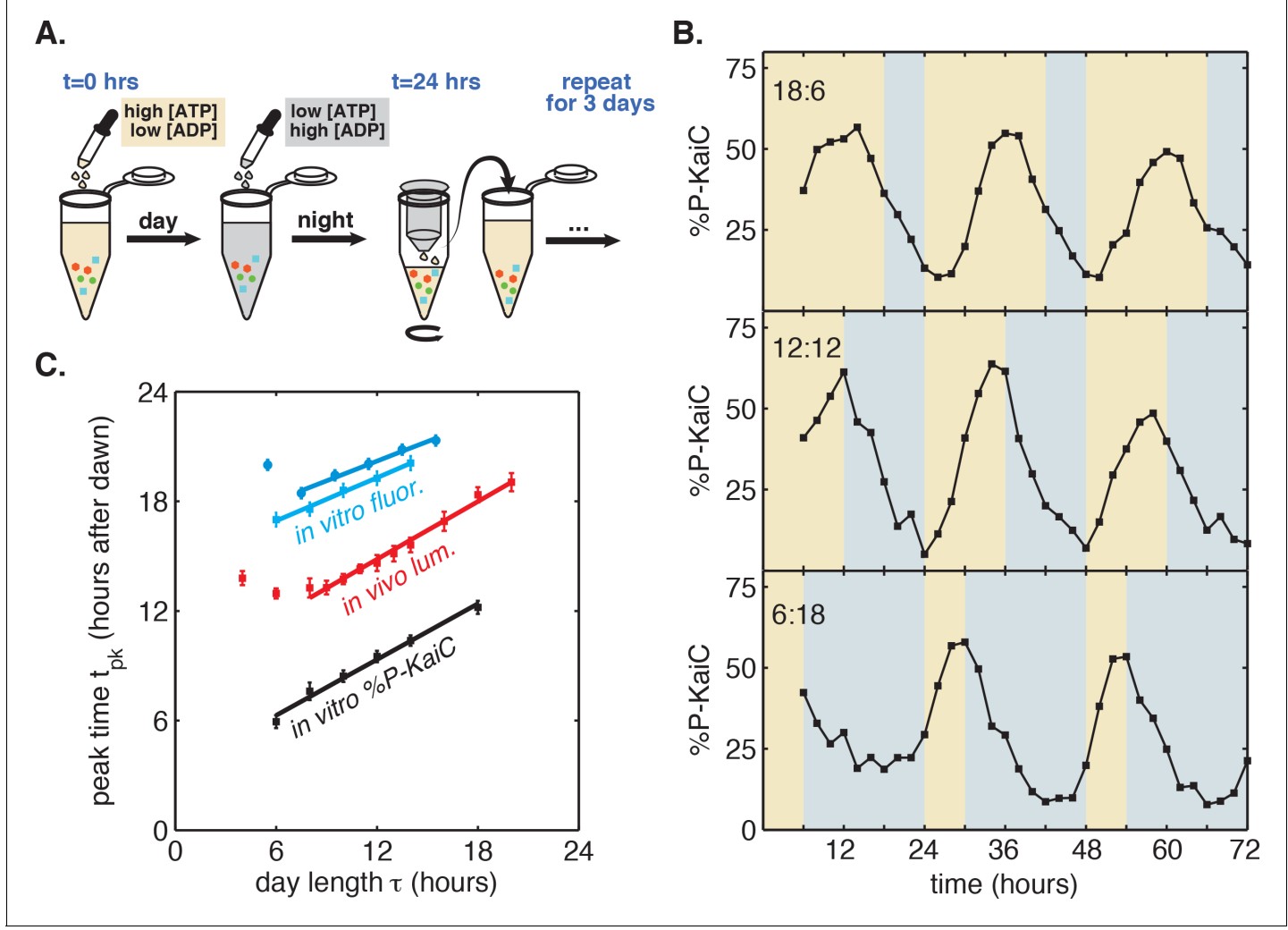

**Figure 2.** Reconstitution of the seasonal clock response in vitro. (**A**) Buffer exchange protocol to simulate metabolic driving of the clock. To mimic daytime in vitro, purified Kai proteins (*green, blue and red symbols*) were incubated in 'day' reaction buffer containing ATP. ADP was added to mimic nightfall ([ATP]/([ATP]+[ADP]) ≈ 25%). At simulated dawn, reactions were returned to 'day' buffer via buffer exchange. (**B**) Example traces of KaiC phosphorylation rhythm from in vitro reactions mimicking LD 18:6 (*top*), LD 12:12 (*middle*), and LD 6:18 (*bottom*). (**C**) Phase of KaiABC oscillation scales linearly with simulated day length (time spent in 'day' buffer), as assessed by the peak time of KaiC phosphorylation (*black squares*) or peak time of fluorescence polarization (*cyan squares and circles*, for two replicates) of fluorescently labeled KaiB. Peak times of fluorescence polarization were estimated from sinusoidal fits to oscillations recorded in free-running conditions after entraining the oscillator with three metabolic cycles. Peak time of %P-KaiC was estimated by fitting sinusoids to KaiC phosphorylation time series from the third day of reactions. Error bars represent uncertainty of fit phase from sinusoidal regression. Lines of best fit were determined by linear regression (*cyan squares: m = 0.39 ± 0.06, cyan circles: m = 0.36 ± 0.04, black: m = 0.51 ± 0.04*). In vivo data (from ***Figure 1C***) is plotted in *red*. Scaling of entrained phase was measured once via KaiC phosphorylation analysis (*black*) and twice using the fluorescence polarization probe (*cyan squares and circles*) with an independent preparation of proteins.

The following source data and figure supplement are available for figure 2:

**Source data 1.** Source data for ***Figure 2B***.

**Source data 2.** Source data for ***Figure 2C***.

**Source data 3.** Source data for ***Figure 2—figure supplement 1***.

**Figure supplement 1.** Validation of KaiB fluorescence polarization reporter against KaiC phosphorylation rhythm.

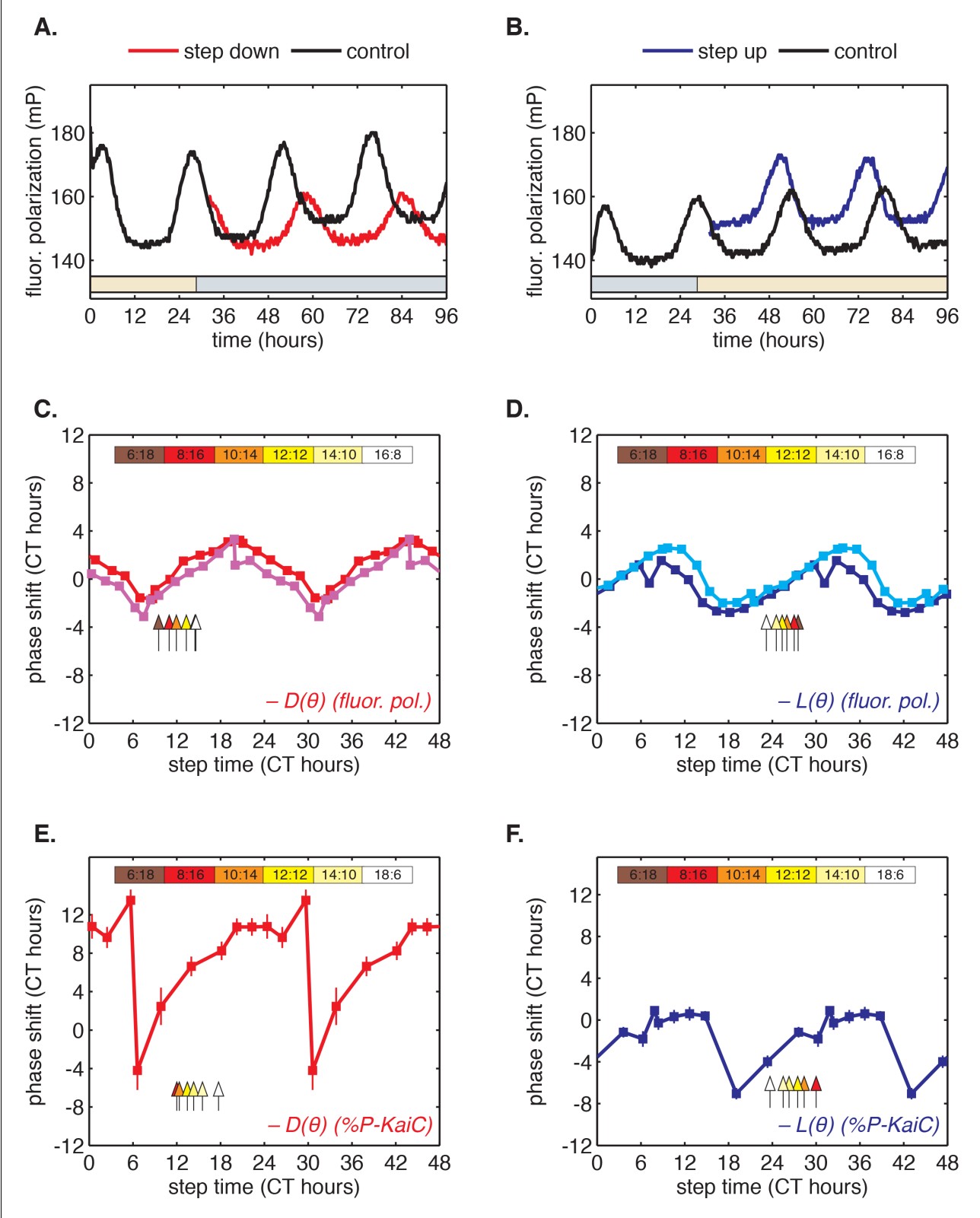

**Figure 3.** Clock responses to metabolic steps mimicking dawn (step up) and dusk (step down). (A) Phase shift in fluorescence polarization (*red curve*) caused by a shift to a buffer that mimics the nucleotide pool at night ([ATP]/([ATP]+[ADP]) ≈ 25%, *gray bar*). The control reaction remained in the original buffer (*black curve*). (B) Phase shift in fluorescence polarization (*blue curve*) caused by a shift from night buffer back to day buffer ([ATP]/([ATP] +[ADP]) ≈ 100%, *beige bar*). (C, D) Summary of phase shifts caused by metabolic step-down (C) or step-up (D) perturbations throughout the clock

*Figure 3 continued on next page*

*Figure 3 continued*

cycle. Simulated day-night or night-day steps were administered as in (A) and (B). Different colors represent independent measurements. To estimate the phase of each reaction, trajectories were fit to sinusoids. Phase shifts were determined relative to the respective control reactions. The times at which buffer steps were administered were converted to circadian time (CT 0 corresponds to the estimated trough of KaiC phosphorylation based on *Figure 2—figure supplement 1*). Colored arrows indicate clock phases when metabolic shifts occur in entrained conditions. (E, F) Analogs of (C, D) for the gel-based phosphorylation measurements on an independent preparation of Kai proteins. Error bars represent standard deviations calculated by bootstrapping (see Computational methods). Horizontal error bars are smaller than marker widths.

The following source data and figure supplements are available for figure 3:

**Source data 1.** Source data for *Figure 3A–B*.
**Source data 2.** Source data for *Figure 3C–D*.
**Source data 3.** Source data for *Figure 3E–F*.
**Figure supplement 1.** Example calculation of phase shifts in response to metabolic step transitions.
**Figure supplement 2.** Experimentally measured step-response functions predict entrainment to driving periods near 24 hr.

linear, regions of both curves, spanning roughly 6 hr windows near subjective dawn (for $L(\theta)$) and subjective dusk (for $D(\theta)$). Because these regions of $L(\theta)$ and $D(\theta)$ represent phase shifts comparable in magnitude but opposite in sign, we hypothesized that their opposing forces could enable oscillator entrainment to light-dark cycles.

To check this hypothesis, we simulated a simple (phase-only) oscillator that runs at constant frequencies in the light and dark and adjusts its phase in response to dawn and dusk according to the phase shift functions that we measured in *Figure 3C–D* (see Computational methods). This model maps the one-dimensional phase variable from one cycle to the next according to $L(\theta)$ and $D(\theta)$. We analyzed the stability properties of this map and found that the oscillator could entrain with a unique phase to driving periods of within 4–5 hr of its natural circadian period. Outside this range, the oscillator failed to entrain or exhibited more complex dynamics (*Figure 3—figure supplement 2*).

We proceeded to test whether the measured $L(\theta)$ and $D(\theta)$ step-response functions could successfully reproduce the entrainment of the oscillator to repeated 24 hr metabolic cycles of varying day length, shown in *Figure 2*. When subjected to light-dark cycles in simulations, the oscillator model stably entrains to the diurnal schedule within two-to-five cycles (*Figure 4A–B*, *Figure 4—figure supplement 1A*). Importantly, the simulated entrained phase scales linearly with day length with a slope similar to the experimental data, indicating that the driven clock response can indeed be decomposed into a series of step responses to environmental transitions (*Figure 4C*).

As described above, we also measured $L(\theta)$ and $D(\theta)$ step-response functions with a separate preparation of Kai proteins using a gel-based assay to read out the phase of the KaiC phosphorylation rhythm (*Figure 3E–F*). Although the absolute magnitude of the step responses in these measurements was larger, they still predict linear scaling of entrained phase in simulations for day lengths between 6 and 14 hr (*Figure 4—figure supplement 2B*). The discrepancy in magnitude between these measurements may point to differences in sensitivity to input cues between different preparations of Kai proteins, or to a slight perturbative effect of fluorescently labeled KaiB.

We found that simulated entrainment of the phase oscillator model was particularly sensitive to the regions surrounding subjective dusk (for $D(\theta)$) and subjective dawn (for $L(\theta)$) (*Figure 3E–F*, *colored arrows*). Because the fluorescence polarization approach allows us to measure many conditions in an automated way over many days, and thus to disentangle phase shifts from period differences (*Figure 3C–D*, *Figure 3—figure supplement 1*), these higher time resolution measurements better constrain the portions of the response functions critical for entrainment.

Given the observation that oscillator entrainment was sensitive to local shapes of $L(\theta)$ and $D(\theta)$, we wondered if midday tracking requires a specific form of these response functions or if certain generic features are sufficient. Based on our observation that $L(\theta)$ and $D(\theta)$ in the KaiABC system are approximately linear in the regions used during metabolic entrainment, we asked whether linear $L(\theta)$ and $D(\theta)$ would result in a linear dependence of clock phase on day length.

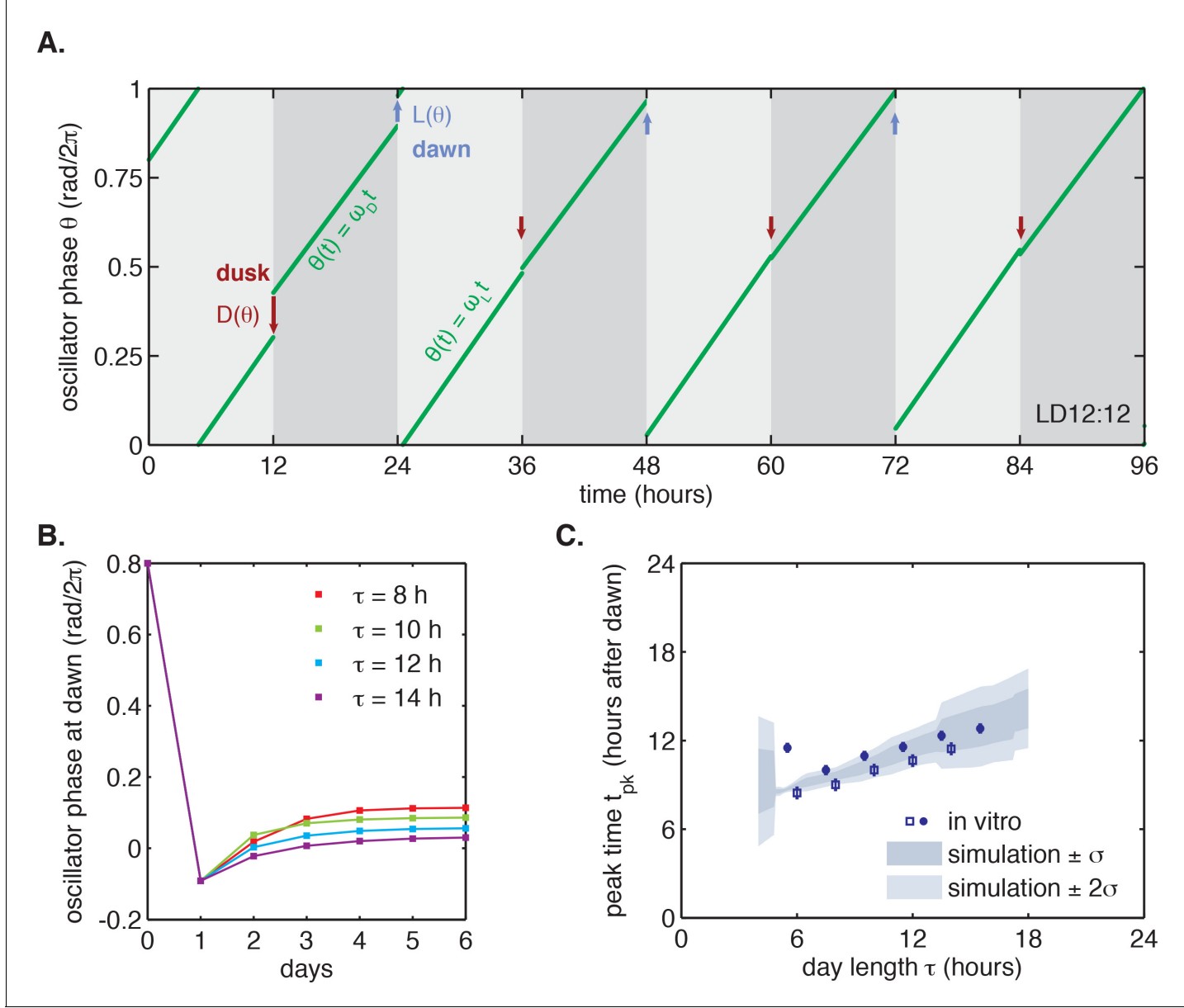

**Figure 4.** Entrainment of the phase oscillator model to a driving cycle. (**A**) Schematic of a phase-only oscillator that responds to dawn and dusk with instantaneous phase shifts. The oscillator runs at constant velocities $\omega_L$ during the day and $\omega_D$ at night (*green lines*), except for dawn and dusk where sudden shifts occur (*red and blue arrows*). This simulation illustrates entrainment to a LD 12:12 cycle for an oscillator with $\omega_L$ (1/23.7 hr), $\omega_D$ (1/25.7 hr), and $L(\theta)$ and $D(\theta)$ as experimentally measured for the KaiABC oscillator (see main text, Computational Methods). Refer to ***Figure 4—figure supplement 1B*** for illustrations of $L(\theta)$ and $D(\theta)$ used in this simulation. (**B**) Simulated approach to stable entrainment in the model (as in ***Figure 4A***) for light-dark cycles of different day length ($\tau$ = 8–14 hr). (**C**) Simulated seasonal response of an oscillator that responds rapidly to light-dark cues according to the phase shift functions in ***Figure 3C*** (see text). In simulations, $t_{pk}$ is defined as the time when oscillator phase $\theta$ equals $\pi$ rad (0.5 cycles), corresponding to the peak of KaiC phosphorylation. Shaded areas correspond to standard deviations of entrainment simulations using the four possible combinations of $L$ and $D$ functions shown in ***Figure 3(C–D)***. *Blue squares and circles* indicate experimentally determined entrained phases measured using the fluorescence polarization reporter in ***Figure 2C***. Peak times in polarization data were converted to equivalent peak KaiC phosphorylation times using the measured phase offset for the polarization reporter (***Figure 2—figure supplement 1***). Error bars on in vitro data show uncertainty of fit phase.

The following source data and figure supplements are available for figure 4:

**Source data 1.** Source data for ***Figure 4C***.
**Source data 2.** Source data for ***Figure 4—figure supplement 2***.

*Figure 4 continued on next page*

*Figure 4 continued*

**Figure supplement 1.** Example simulation of a phase oscillator governed by one set of experimentally determined $L(\theta)$ and $D(\theta)$ functions and subjected to a driving cycle.

**Figure supplement 2.** Simulations of seasonal entrainment for a phase oscillator driven by linearized step-response functions ($L_{lin}$ and $D_{lin}$).

**Figure supplement 3.** Dependence of the slope of entrained phase on the slopes of step response functions.

## Linear regions of step-response functions underlie proportional tracking of day length

Consider a phase oscillator driven by light-dark cycles of period $T$ and day length $\tau$ (*Scheme 1*). When the oscillator is entrained (phase-locked) to the light-dark cycle, the oscillator returns to the same state by the end of a full cycle. Starting from an initial phase $\theta_0$, the clock accumulates phase $\omega_L \tau$ during the day, then experiences a phase shift of magnitude $D(\theta_\tau)$ at dusk, accumulates phase $\omega_D(T - \tau)$ at night, and finally responds to dawn with a phase shift $L(\theta_T)$:

| phase $\theta(t)$ | $\theta_0$ | $+D(\theta_\tau)$ | $+L(\theta_T)$ | | |
|---|---|---|---|---|---|
| | | $+\omega_L\tau$ | $+\omega_D(T-\tau)$ | | |
| time $t$ | $0$ | $\tau$ | $T$ | $T+\tau$ | $2T$ |

$$\theta_T = \theta_0 + \omega_L\tau + D(\theta_\tau) + \omega_D(T-\tau) + L(\theta_T)$$

**Scheme 1.** Phase oscillator entrainment.

Here all angles are measured in units of cycles ($1\ \mathrm{cyc} = 2\pi$), and $\omega_L$ and $\omega_D$ are the oscillator frequencies in the light and dark, respectively. For stable entrainment, the effects of $L(\theta)$ at dawn and $D(\theta)$ at dusk must balance the phase accumulated by the oscillator, so that the phase returns to the same point at the end of each cycle.

This expression for the entrained clock phase scales linearly with day length if its derivative with respect to $\tau$ is constant. The simplest way to achieve this condition is if $L(\theta)$ and $D(\theta)$ are themselves linear functions of $\theta$, such that $D(\theta) \approx -d(\theta - \theta_D)$ and $L(\theta) \approx -l(\theta - \theta_L)$ over the relevant range of clock times. If oscillator frequency is the same in light and dark, as is approximately true for the Kai oscillator ($\omega_D/\omega_L = 0.93 \pm 0.01$, see Computational methods), the peak time of the oscillation (measured in hours after dawn) can be expressed as $t_{pk} = m\tau + C$, with the slope $m(l,d) = d(1-l)/(d+l-ld)$ determined by the slopes of the linear $L(\theta)$ and $D(\theta)$ functions. If the day and night oscillator frequencies differ, we still obtain a linear dependence on day length, but with an altered expression for the proportionality constant $m$ (see Appendix 1). This expression for $m$ imposes constraints on values of $l$ and $d$ required for an oscillator to track different portions of the day-night cycle. *Figure 4—figure supplement 3* highlights the requirements on $l$ and $d$ for a midday-tracking clock.

To determine whether $L(\theta)$ and $D(\theta)$ for the Kai oscillator are in line with these mathematical requirements, we examined the linear portions of the step-response functions in *Figure 3*. Indeed, the slopes of $L(\theta)$ and $D(\theta)$ from the fluorescence polarization assay ($l = 0.34 \pm 0.03$, $d = 0.38 \pm 0.05$) predict an $m$ value consistent with our measurements of the entrained in vitro oscillator using the same method ($m(l, d) = 0.34 \pm 0.04$, calculated in the linear model vs. $m = 0.38 \pm 0.07$, measured) (*Figure 4—figure supplement 3*).

Because $L(\theta)$ and $D(\theta)$ are periodic functions on a circle, they cannot be linear everywhere with non-integer slope. However, maintaining the linear scaling of phase with day length only requires that both $L(\theta)$ and $D(\theta)$ be linear over the range of clock times when dawn and dusk occur, respectively, with the nonlinearities required to satisfy periodicity appearing at other times. The exact width of the linear region of $L(\theta)$ and $D(\theta)$ depends on $m$ and the range of day lengths that the oscillator is required to track. For example, an oscillator capable of tracking midday ($m = 0.5$) over a 12 hr range

of day lengths requires that $L(\theta)$ and $D(\theta)$ be linear over at least one quarter of the cycle. This criterion is met by the measured $L(\theta)$ and $D(\theta)$ (**Figures 3C-D**).

To further test whether entrainment of the Kai system can be described by this framework, we returned to our phase oscillator simulations (**Figure 4**). When we replaced our experimentally measured $L(\theta)$ and $D(\theta)$ functions with linear approximations, the simulated oscillator exhibited a similar scaling of entrained phase over a wide range of day lengths (**Figure 4—figure supplement 2A–B**). The linear approximations work because the regions where the step response functions deviate strongly from linearity are avoided in our entrainment simulations (**Figure 4—figure supplement 2C**).

Together, these results indicate that the driven behavior of the cyanobacterial circadian clock in 24 hr cycles can be approximated as a simple oscillator that shifts in response to light-dark transitions with a sensitivity that varies linearly with phase. To test whether this linear mathematical framework holds more generally and to understand how the cyanobacterial clock responds to a broader range of environments, we sought to measure clock response to a variety of external conditions with both single and repeated light-dark transitions.

## The behavior of the driven circadian clock across diverse conditions can be collapsed to a simple mathematical representation

The phase oscillator model described above was inspired by the observation of linear scaling of clock phase in 24 hr days with various day lengths. The key model assumptions are (i) a unique clock cycle in the light and in the dark, (ii) rapid relaxation from one cycle to another when conditions change, and (iii) sensitivity to environmental changes that varies linearly with clock time. While these assumptions hold, the model should be capable of describing the behavior of a biological clock in arbitrary fluctuating environments. For example, the response to a single dark pulse can be decomposed into sequential step-down and step-up responses.

To test the range of validity of this mathematical model, we used our LED array system (**Figure 1A**) to collect data on *S. elongatus* clock function in response to dark pulses administered at different clock times. This corresponds to a classical phase response curve (PRC) analysis, a commonly used tool in circadian biology for characterizing the response of a biological clock to perturbations. We also probed clock responses to dark pulses of varying lengths (a so-called 'wedge' analysis), and to repeated light-dark cycles with periods different from 24 hr. Our phase oscillator model predicts that the quantitative response to these various perturbations should all be related through the step-up and step-down functions (see Appendix 1). One specific prediction of the model is that changing the time at which a dark pulse begins and the duration of a dark pulse should have separable linear effects on the oscillator, which would manifest as linear curves in phase response and wedge analyses. These experimental data and a fit of our linear model to the results are presented in **Figure 5**. The lines of best fit in **Figure 5A–C** were obtained from a global fit to all of the datasets, with two fitting parameters that fix the slopes of the regression lines across all conditions (Appendix 1). The overall agreement between the model and data suggest that limited data on clock response can be successfully extrapolated to other conditions using this approach.

To understand the implications of these results, consider the phase response curve (PRC) for a 12 hr dark pulse (**Figure 5A**). Although phase response analyses frequently separate PRCs into regions of low sensitivity to perturbation ('dead zones') followed by highly nonlinear regions with large phase shifts, our model suggests a different interpretation (**Pattanayak et al., 2014**; **Johnson, 1999**; **Schmitz et al., 2000**; **Pfeuty et al., 2011**). Clock response is weakest at times when nightfall is expected ($t$ = 36 and 60 hr) and increases gradually with clock time. The slowly changing regions of the PRC are well described by line segments with the same slope, in agreement with our model based on linear step-response functions $L(\theta)$ and $D(\theta)$. The 'breakpoint' in the PRC near 50 hr is a consequence of the fact that these periodic step-response functions cannot be linear everywhere with non-integer slope, as discussed earlier.

In other words, a circadian clock capable of tracking midday over a wide range of day lengths is expected to have a PRC that is approximately linear over many hours except for a narrow region of rapid change that is required to satisfy periodicity. Indeed, in simulations where $L(\theta)$ and $D(\theta)$ are linear everywhere except for a discontinuous jump, the PRC for a long dark pulse appears as a straight line except for a single breakpoint (**Figure 5—figure supplement 1**). Near the breakpoint, small changes in the timing of darkness result in large differences in phase shifts, and we would expect to

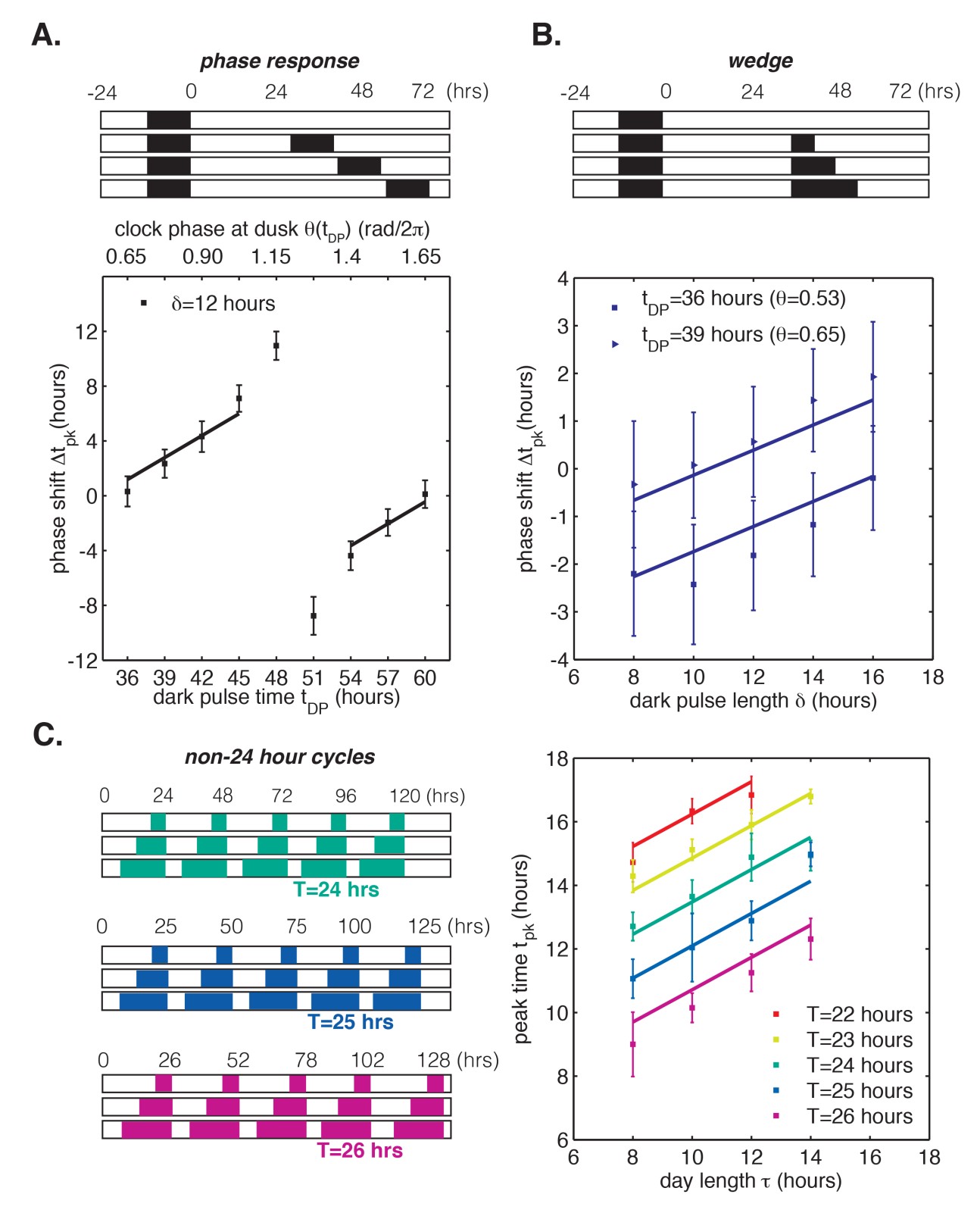

**Figure 5.** Phase oscillator model with linear phase shift functions predicts entrainment of the cyanobacterial clock to different light-dark (LD) patterns. In all panels, error bars represent standard deviations (n = 4–8 technical replicates per point). Lines are fit globally to all three datasets in (A)-(C). See Computational methods for details. (A) Phase resetting analysis. Phase shifts of bioluminescence rhythm ($P_{kaiBC}$::$luxAB$) due to 12 hr dark pulses ($\delta$ = 12 hr) administered throughout the circadian cycle. The experimental protocol is represented schematically above the graph. Cells were exposed to one

*Figure 5 continued on next page*

*Figure 5 continued*

12 hr dark pulse and released into constant light; 12 hr dark pulses were administered at the indicated times. $\theta(t_{DP})$ is the clock phase at beginning of the dark pulse, with $\theta = 0$ defined as clock phase at the trough of the bioluminescence rhythm. (B) Wedge analysis. Phase shifts of bioluminescence rhythm ($P_{kaiBC}::luxAB$) due to dark pulses of varied length ($\delta$ = 8–16 hr) administered near subjective dusk (36 or 39 hr after an initial 12 hr dark pulse). Clock phases at the beginning of the dark pulse are listed in parentheses; $\theta = 0$ is defined as clock phase at the trough of the bioluminescence rhythm. The experimental protocol is represented schematically above the graph. (C) Seasonal response in non-24 hour environmental cycles. Cells were grown in LD cycles with period T = 22–26 hr and day length $\tau$ = 8–14 hr (see schematic on left). After five entraining cycles, cells were released into LL and the phase of the circadian rhythm was estimated by sinusoidal regression.

The following source data and figure supplement are available for figure 5:

**Source data 1.** Source data for *Figure 5A–C*.
**Figure supplement 1.** Simulation of a phase-resetting curve.

find this region of the PRC at a clock time when such a perturbation is least likely to occur naturally. In the dark-pulse PRC for *S. elongatus*, the breakpoint near 50 hr occurs when prolonged darkness is improbable (*Figure 5A*). The maximal phase shift in a PRC can be a useful tool to characterize clock mutants, but our analysis highlights that these regions are unlikely to be experienced in natural conditions and may exist only as consequences of periodicity constraints.

## A geometric interpretation of oscillator response to varying day length

Finally, we asked how step response functions with linearly increasing sensitivity are related to the mathematical structure of the oscillator and ultimately to the underlying molecular mechanism. Do step response functions with these properties arise generically, or do they require fine-tuned choices of parameters? To address these issues, we considered the simplest possible dynamical representation that allows for distinct day and night clock cycles. In this model, the clock cycle during the day is represented by a circular limit cycle with unit radius, and the clock time is defined by the angular coordinate of the oscillator on this limit cycle (*Figure 6A*). The effect of darkness is to deform the limit cycle, transforming the daytime orbit to a nighttime orbit. For simplicity, we assume this night cycle lies in the same plane as the day cycle and is also circular, but may be offset relative to the day cycle and have a distinct radius (*Figure 6A*).

After a transition from one condition to the other, the system state must evolve to the new limit cycle. We suppose that each cycle is very strongly radially attracting. That is, when the system is in a state off the limit cycle, it is rapidly pulled to the closest point on the cycle. Under these conditions, the step transitions from one cycle to another are determined purely by geometry (*Figure 6A*), and we can connect this picture with the phase-only description we used to analyze the experimental data. Here, linearly increasing sensitivity of a step response has a simple geometric interpretation: when the two cycles are displaced from each other, a step transition maps an arc of one cycle onto an arc of the other cycle that subtends a smaller angle than the original (*Figure 6B*). Thus, step transitions compress or expand angular distance when mapping one circle onto another. The slopes of the step response functions are given by the compression factor in this mapping (*Figure 6C–D*, *Figure 6—figure supplement 1*; see Appendix 1).

To determine how the entrained clock phase depends on geometry in this model, we simulated step transitions and then calculated the slope $m$ of clock phase versus day length (*Figure 6E*). This calculation indicates that midday tracking ($m \approx 0.5$) requires that the separation between the light and dark cycles is comparable to the radius of the cycle ($R \approx X$). This requirement can be understood intuitively by considering three cases where orbits have the same center-to-center separation (log $X = 0.5$) but different sizes (*Figure 6—figure supplement 1*). Dusk transitions are strongly resetting for this choice of $X$, but the strength of dawn resetting varies with the relative size of the orbits. Dawn-tracking entrainment results when the night orbit is much smaller than the day orbit ($l \approx 1$) and dusk-tracking entrainment results when the night orbit is much larger ($l \approx 0$, $d \approx 1$), in accord with how $m$ varies as a function of $l$ and $d$ (*Figure 4—figure supplement 3*). When the size of both orbits is similar to their center separation, the oscillator can track intermediate phases, such as midday.

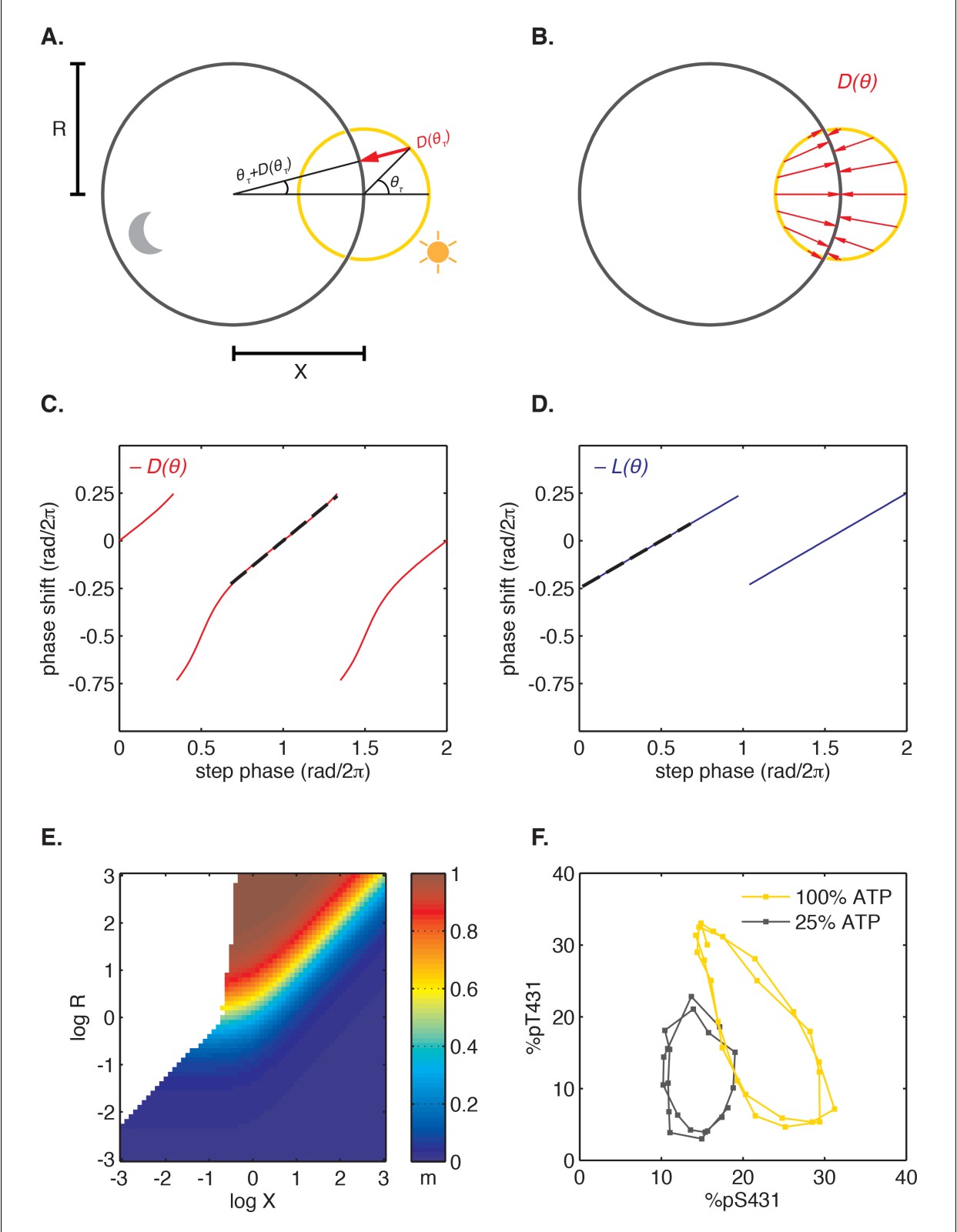

**Figure 6.** Nearly linear step response functions can arise from the relative geometry of day and night limit cycles. (A) Geometric model of oscillator phase resetting. During the day, the oscillator runs with constant angular velocity along the daytime orbit (*yellow*), which has unit radius and is centered at the origin. At dusk, the oscillator transits to the nighttime orbit (*black*), which has radius $R$ and is displaced from the daytime orbit by $X$ units. In the limit where the nighttime orbit is strongly radially attracting, we can approximate oscillator response to the light-dark transition ($D(\theta_\tau)$, *red arrow*) as an

*Figure 6 continued on next page*

*Figure 6 continued*

instantaneous jump from phase $\theta_\tau$ on the daytime orbit toward the center of the nighttime cycle, resulting in phase $\theta_\tau + D(\theta_\tau)$ on the night orbit. (B) Simulation of oscillator phase shifts due to light-dark transitions at different phases on the day orbit (red arrows) for $R = 2$, $X = 2$. For geometries with $X \approx R$, phase angles on the day orbit are compressed to an arc on the night limit cycle that subtends a smaller angle. See Computational methods for calculation details. (C and D) Simulations of $L(\theta)$ and $D(\theta)$ step response functions arising from the geometric arrangement of day and night cycles in (B). Linear regions of $L(\theta)$ and $D(\theta)$ are marked with black dashes. See Computational methods and Appendix 1 for calculation details. (E) Heat map of the slope $m$ of the approximately linear relationship between entrained phase and day length, plotted as a function of $X$ and $R$. In white regions, the oscillator does not entrain stably or the oscillator does not show linear scaling of phase with day length. Slope determined from simulations of oscillator entrainment to 24 hr driving cycles of day length $\tau = 6$–18 hr. See Computational methods for details. (F) Limit cycles traversed by the KaiABC oscillator in vitro in metabolic conditions mimicking day (yellow, [ATP]/([ATP]+[ADP]) $\approx$ 100%) and night (black, [ATP]/([ATP]+[ADP]) $\approx$ 25%). Oscillations in KaiC phosphorylation on Ser431 and Thr432 are replotted from data in *Phong et al. (2013)*.

The following figure supplements are available for figure 6:

**Figure supplement 1.** Illustrations of limit cycle geometries that give rise to step-response functions $L(\theta)$ and $D(\theta)$ with different slopes, resulting in dusk-, dawn-, or midday-tracking entrainment.

**Figure supplement 2.** The relative size ($R$) and center-to-center distance ($X$) of day and night limit cycles are major determinants of entrained behavior.

**Figure supplement 3.** Interpretation of $m$, the slope of the approximately linear relationship between entrained phase and day length.

Although this model makes a simple connection between attractor geometry and entrainment, it also makes a number of simplifying assumptions about oscillator dynamics that may not hold true for real biological clocks, such as instant transitions between cycles, perfectly circular orbits and constant angular frequencies along each cycle. To test the consequences of relaxing these assumptions, we used a dynamical model where the evolution of the system is described explicitly (see Computational methods). Our simulations in *Figure 6—figure supplement 2* show that in these more complicated scenarios the geometric arrangement of day and night cycles remains a key determinant of the slope of entrained clock phase as a function of day length. Indeed, in all cases we studied, midday tracking was only possible for geometries where the center-to-center distance between the day and night cycles was comparable to the radius of the larger orbit ($R \approx X > 1$).

This dynamical systems perspective allows us to reframe conditions on the underlying biochemical mechanisms that can produce the observed midday tracking behavior. Changes in the external environment caused by transitions between night and day should affect the oscillator in such a way that the period remains close to 24 hr, but that the limit cycle is shifted by an amount comparable to its radius. The relative geometry of the two limit cycles, which is determined by the mechanisms that couple the environment to the clock, must be fine-tuned to give a specific slope for the entrained phase. Indeed, when we plotted experimentally determined orbits of the purified KaiABC oscillator on axes showing the extent of phosphorylation of two key sites, we observed an arrangement similar to the expected geometry ($R \approx X$) using nucleotide conditions that simulate either day or night (*Figure 6F*) (*Rust et al., 2007*; *Phong et al., 2013*).

## Discussion

Although circadian clocks are defined in part by their ability to continue to cycle in constant environments, the defects associated with clock mutants are often most apparent when organisms are faced with fluctuating environments (*Ouyang, 1998*; *Woelfle et al., 2004*; *Pittendrigh, 1972*; *Spoelstra et al., 2016*; *Pittendrigh and Minis, 1972*). Thus, an important challenge is to understand how biological clocks respond to the cycling environments found in nature, and how they function to appropriately schedule gene expression and behavior.

For most organisms, there is an asymmetry between day and night, in terms of food availability, predation risk, etc., so that the need to carry out certain activities diurnally or nocturnally presents changing demands as the length of the day varies throughout the year. The situation is especially dramatic for cyanobacteria because there is an extreme metabolic contrast between day and night. We found that *S. elongatus* contends with these challenges using a clock that tracks the middle of the day.

The ability of circadian clocks to keep track of the phase of the light-dark cycle has been long recognized in plants, insects, rodents and higher mammals (*de Montaigu et al., 2015*; *Daan et al., 2001*; *Edwards et al., 2010*; *Hut et al., 1999*, *Hut et al., 2013*; *Wehr, 2001*). Although molecular mechanisms that give rise to these entrainment behaviors are still being uncovered, analysis of circadian clock models has found that the presence of multiple feedback loops in complex clocks determines the number of points in the driving cycle that the oscillator can track simultaneously, by allowing different internal phase relationships between the clock components (*Rand et al., 2004*). For example, the multi-feedback loop clocks in plants are able to track phases of both dawn and dusk (*Edwards et al., 2010*).

Consistent with this picture, we find that the core circadian oscillator in cyanobacteria, which relies on a single posttranslational feedback loop, keeps track of a single phase—the midpoint of the day portion of the cycle, a property described in our mathematical framework as a linear scaling of entrained phase with day length with slope $m \approx 0.5$. Why might keeping track of midday be useful for a photosynthetic organism with a simple clock? Clock-controlled gene expression in *S. elongatus* tends to be bimodal, with most genes falling into subjective dawn or dusk classes. Because the biosynthetic capacity of *S. elongatus* is severely limited in darkness, the midday tracking effect we describe here could be a mechanism to ensure that biosynthetic resources are partitioned in a balanced way between the dawn and dusk genes, even as the day length changes with the seasons (*Figure 6—figure supplement 3*). In particular, clock-driven transcription in this organism has been shown to implement a switch between anabolic and catabolic carbon metabolism, suggesting that a role for the midday tracking we observe here is to ensure balanced growth by timing this switch appropriately in days of different length (*Diamond et al., 2015*).

The ability to reconstitute this effect in vitro by delivering metabolic pulses to the purified Kai proteins indicates that midday tracking is not necessarily achieved through additional feedback mechanisms in the cell, but appears to be a property of the clock proteins themselves. The purified clock responds to metabolic steps with phase shifts that are linear functions of the previous phase. The slopes of these response functions are presumably tuned to give an appropriately entrained clock phase. Notably, linear responses have also been observed for the Kai oscillator following temperature steps, suggesting that this is a general reaction of the system to inputs (*Yoshida et al., 2009*).

The mathematical framework that we describe here has deep similarities to the theory of nonparametric entrainment developed by Colin Pittendrigh (*Pittendrigh and Daan, 1976*; *Daan, 2000*; *Pittendrigh and Minis, 1964*). His work motivated a theory of entrainment to diurnal cycles mediated by instantaneous phase shifts at dawn and dusk, which can be summarized by a phase response curve. Daan and Pittendrigh suggested that the ability of the clock to track specific phases of the day-night cycle in different seasons depends on the shape of the phase response curve as well as the difference between the free-running period of the clock and the period of the day-night cycle (*Pittendrigh and Daan, 1976*; *Johnson et al., 2003*). Our decomposition of driven behavior of the KaiABC oscillator into individual step responses is in the spirit of this classic paradigm.

We note that a mismatch between the free-running clock period and the external cycle is not required for stable entrainment in our phase oscillator model (see Appendix 1). Instead, entrainment can be achieved from the opposing effects of the step-up and step-down phase shifts, which occur at different clock times in days of different length. The step-response curves underlying entrainment in our model are nonlinear functions of clock phase, but they can be successfully approximated by linear functions over the interval of clock phases used during entrainment. Our simulations suggest that the slopes of these locally linear functions are key determinants of entrained phase, along with changes in clock period in daytime and nighttime conditions. Successful prediction of how the in vitro oscillator entrains to rhythmic environments is due in large part to the ability to map out the step-response curves with high temporal resolution. In this study, we achieve this by measuring free-running oscillations in an automated way using a fluorescence polarization probe.

Even though there are likely other effects at play in natural environments, as long as the system can be described by fast relaxation back to distinct limit cycles in day and night, instantaneous step-response descriptions of the kind used here should be applicable. The data reduction achieved in our linear model holds the promise of predicting the behavior of the circadian rhythms in many time-varying environments from a minimal data set that characterizes oscillator response, and may be applicable to clocks in many organisms.

Although the biochemistry of even the simplest circadian clock is complex, our analysis suggests that a key determinant of entrained behavior is how the clock limit cycle is deformed by coupling to the environment. Viewed in this way, the entrainment properties of circadian oscillators arise from simple features of the geometry of the limit cycle attractor that could be measured in any organism. The concept that the phase-shifting of oscillators can be studied in terms of their geometric properties was initially developed by Winfree (*Winfree, 1973*). In this context, it would be informative to analyze the geometric arrangement of day and night limit cycles of clock gene transcripts in other species. An important goal for the future is to understand how the biochemical properties of the clock components in cyanobacteria and other organisms achieve the effect of shifting the limit cycle without changing the period, allowing us to use dynamical systems theory to bridge the gap between molecular detail and systems-level clock phenotypes.

# Materials and methods

## Experimental methods

### Cyanobacterial strains and culture conditions

Two strains of *S. elongatus* PCC 7942 were used for this study. AMC1300 is a wild-type derivative carrying a bacterial luciferase bioluminescent reporter of *kaiBC* expression. AMC1300 carries *PkaiBC::luxAB* at NS1 and *PpsbAI::luxCDE* at NS2, which enables the cells to produce the luciferase enzyme and the long-chain aldehyde substrate for the luminescent reaction (*Chen et al., 2009*). AMC408 carries a *purF* reporter (*PpurF::luxAB* at NS2, *PpsbAI::luxCDE* at NS1) (*Nair et al., 2002*). Prior to experimental measurements, all cultures were grown in test tubes in BG11M liquid medium at 30°C with shaking under cool white fluorescent bulbs ($\approx 60$ µmol photons $m^{-2}$ $s^{-1}$, Philips AltoII, Amsterdam, Netherlands).

### Creating custom light-dark environments using LED arrays

To simulate different light-dark cycles, we used custom-built red LED arrays in a 96-well format (LEDs from superbrightleds.com, St. Louis, MO, cat. no. RL5-R12008; 96-well plates from Corning, Corning, NY, cat. no. 3916). The LEDs were mounted into a hollowed-out 96-well plate and the tails of the LEDs were soldered into a circuit board, where they were wired in parallel in groups of four to the analog inputs of an Arduino Mega 2560 microcontroller. Another hollow 96-well plate was glued to the bottom of the LED-carrying plate, beneath the LEDs, in order to create a light baffle and prevent light leakage between the wells. These devices were placed $\approx 2$ mm above a black 96-well plate containing cells growing on BG11-agar, such that every well of the growth plate received illumination from a single LED ($\approx 1.8$ cm between LED and agar surface). The growing plate was sealed with transparent film, with holes punctured above each well to provide aeration. Custom Arduino scripts were written to administer appropriate light-dark schedules to cells in the growth plates. Every well received the same light intensity (1.8 V across each LED, $\approx 200$ µmol photons $m^{-2}$ $s^{-1}$) in the light portion of the day. The temperature of the agar was $31.6 \pm 0.8$ °C underneath LEDs that were turned on and $28.9 \pm 0.3$ °C under LEDs that were off (mean $\pm$ standard deviation of 6–10 wells).

### Monitoring gene expression in vivo using bioluminescent reporters

Cells were grown in test tubes until OD 0.5–0.8, as described above, and diluted to OD 0.2 immediately prior to experiment. A black 96-well plate was filled (200 µL/well) with BG11M-agar (15 g/L agar) supplemented with sodium thiosulfate (1 mM) and HEPES (20 mM, pH 8.0). When the BG11M-agar mixture cooled to room temperature, 35 µL of cells from growing culture at OD 0.2 were added to each well of the plate. The plate was sealed with transparent UniSeal (GE HealthCare Life Sciences, Pittsburgh, PA, cat no. 7704–0001), holes were punched above each well using a 26G½ needle (BD, Franklin Lakes, NJ, cat. no. 305111), and the plate was placed underneath an LED array. Bioluminescence from luciferase reporters was measured every 30 min using a TopCount scintillation counter (PerkinElmer, Boson, MA). Each well of a 96-well plate was read for 1 s per measurement.

## KaiABC in vitro reactions

KaiA, KaiB, and KaiC were recombinantly expressed and purified as previously described (*Lin et al., 2014*), although the anion exchange chromatography purification step was omitted in the preparation of KaiC used for fluorescence polarization experiments. Protein concentration was measured via Bradford Assay Kit using bovine serum albumin (BSA) as a standard (BioRad, Hercules, CA). For experiments relying on SDS-PAGE analysis of KaiC phosphorylation, KaiABC proteins were mixed in master reaction buffer (20 mM Tris [pH 8], 150 mM NaCl, 5 mM MgCl$_2$, 10% glycerol, 0.5 mM EDTA) supplemented with a mixture of ATP and ADP (day buffer: 2 mM ATP, night buffer: 2 mM ATP, 7.5 mM ADP). All reactions were incubated at 31°C. To mimic light-to-dark transitions, ADP was added to appropriate reactions to 7.5 mM final [ADP]. To mimic dark-to-light transitions, the reactions were passed through Zeba desalting columns (7 MW kDa cutoff, ThermoFisher, Waltham, MA) equilibrated in day buffer. Because every buffer exchange step dilutes the proteins by about 10%, the reactions were prepared at 3× standard protein concentration (10.5 µM KaiB and KaiC, 4.5 µM KaiA). KaiC phosphorylation was assayed by SDS-PAGE and quantified by densitometry, as previously described (*Phong et al., 2013*).

In the cases where oscillations were read out by monitoring fluorescence polarization, KaiABC proteins were mixed at 3× standard protein concentration in the master reaction buffer supplemented with a mixture of ATP and ADP (day buffer: 2.5 mM ATP, night buffer: 2.5 mM ATP, 7.5 mM ADP). All reactions were incubated at 31°C. Day-to-night transitions were mimicked by addition of 7.5 mM ADP (final), and night-to-day transitions were mimicked by passing the reactions through Zeba desalting columns twice.

For step-up perturbations shown in blue in *Figure 3D*, buffer exchange steps were administered every 2 hr over a 24 hr interval, and phase shifts were measured relative to an unperturbed control reaction. For every other experiment in *Figure 3C–D*, buffer exchange or ADP addition steps were performed every 2 hr over a 12 hr interval on two out-of-phase reactions, prepared as follows. A master mix containing proteins and appropriate nucleotides was split into two tubes, which were flash-frozen in liquid nitrogen immediately after mixing and stored at −80°C. To prepare out-of-phase reactions, one of the two tubes was thawed in a 30°C water bath 12 hr later than the other.

After all buffer exchanges were completed, the reactions were supplemented with fluorescently labeled KaiB (0.2 µM final) and transferred to the plate reader.

## Preparation and labeling of fluorescently tagged KaiB

We introduced a K25C mutation in KaiB using site-directed mutagenesis of the pMR0019 plasmid carrying a 6xHis-PSP-KaiB$^{WT}$ construct in the pET47b(+) backbone. KaiB$^{K25C}$ was expressed in BL-21 (DE3) *E. coli* by overnight induction with IPTG at 18°C and purified following the standard protocol (*Lin et al., 2014*). Labeling with 6-iodoacetofluorescein (6-IAF) was performed as previously described (*Chang et al., 2012*) with minor modifications. Briefly, 130 µL of KaiB$^{K25C}$ stock (50–100 µM) was buffer-exchanged into labeling buffer (20 mM Tris, 1 mM TCEP, pH 7.0–7.5) using a Zeba desalting column (7 MW kDa cutoff). A freshly prepared solution of 6-IAF (Life Technologies Corp., Grand Island, NY) in DMSO was added to the protein solution in 10-fold molar excess, and the mixture was incubated overnight at 4°C in the dark. The labeling reaction was quenched by addition of DTT in approximately 10-fold molar excess relative to 6-IAF. Unincorporated dye was removed through three rounds of fivefold dilution in reaction buffer and subsequent concentration using a centrifugal filter (10 kDa cutoff, Amicon, EMD Millipore, Billerica, MA). Concentration of final fluorescein-labeled KaiB$^{K25C}$ solution was determined by Bradford assay.

## Monitoring fluorescence polarization rhythms using labeled KaiB

Oscillations in KaiABC reaction mixtures supplemented with fluorescently labeled KaiB$^{K25C}$ (10.5 µM KaiB and KaiC, 4.5 µM KaiA, 0.2 µM KaiB-fluorescein) were monitored on an Infinite F500 plate reader equipped with a fluorescence polarization module (Tecan Trading AG, Switzerland). At least 30 min prior to measurement, the built-in heating module was turned on to warm the instrument and a black 384 well-plate (Greiner Bio-One, Monroe, NC, cat. no. 781900) was loaded into the plate reader. For the metabolic entrainment experiment shown in blue in *Figure 2—figure supplement 1A* and the step-up experiment shown in blue in *Figure 3D*, the plate reader temperature was 28°C; for all other experiments, the plate reader temperature was 31°C. Reactions were quickly transferred

onto the pre-warmed plate (20–35 µL/well) and one to three wells were filled with master reaction buffer. The plate was sealed with a polyethylene silicone plate sealer (Nunc, ThermoFisher cat. no. 235307) and returned to the instrument. Fluorescence polarization of wells of interest (exc. 485 nm, 20 nm bandpass; em. 535 nm, 25 nm bandpass; dichroic 510 nm.) was read out every 15 min using a script created in iControl software (v. 1.12, Tecan Trading AG). Wells containing reaction buffer only were used as blanks, and the G factor was calibrated such that a solution of free fluorescein in reaction buffer produced a reading of $\approx 20$ mP.

## Data analysis

All oscillating trajectories were fit to sinusoids using optimization routines written in MATLAB (Mathworks, Inc.) (RRID: SCR_001622). See Computational methods for detailed fitting procedures and descriptions of simulations. Computational pipelines used for analysis and simulations have been deposited to GitHub at https://github.com/euleip/Simulations_and_analysis_pipelines_github_repo (*Leypunskiy, 2017*). A copy is archived at https://github.com/elifesciences-publications/Simulations_and_analysis_pipelines_github_repo).

# Computational methods

## Optimization and curve-fitting routines

All nonlinear fitting procedures were written in MATLAB using the lsqnonlin() or nlinfit() routines. Linear regressions were performed using the polyfit() function. Uncertainties around the best-fit slopes were evaluated using standard formulas for linear regression with known errors in dependent variables (*Press et al., 1992*). For in vivo experiments, these errors were computed as standard errors of the mean from replicate measurements; for in vitro measurements in *Figure 2*, the errors in fit phases were computed from the curvature of the cost function at the optimum using the nlpredci() function in MATLAB. Where indicated, 95% confidence intervals were also estimated by nlpredci().

## Normalization of bioluminescent reporter traces

Prior to fitting, bioluminescence trajectories were normalized to zero mean and unit standard deviation over the fitting interval, unless specified otherwise. In all analyses, we discarded data from the first 2.5 hr after lights-on to avoid masking effects.

## Estimation of clock phase in vivo in light-dark cycles

The bacterial luciferase reporter system exhibits transient masking effects following dark-to-light transitions. To overcome this issue in determining the phase of gene expression in light-dark cycles of varied day length, we employed a drive-and-release strategy. In this approach (illustrated for LD 8:16, LD 12:12, and LD 16:8 in *Figure 1A*), cells were first entrained to a given diurnal schedule and then released into constant light for several days to determine the peak times of the entrained rhythm.

We relied on two approaches to calculate the peak times of normalized bioluminescence trajectories recorded after release into constant light: (i) we fit sinusoids to 48 hr segments of the trajectories in constant light, and (ii) we locally fit parabolas near the maxima of the trajectories. The advantage of sinusoidal fitting is that it captures phase information for the entire waveform; on the other hand, local parabolic fitting allows for a precise determination of the time of an individual peak without influence from others. In practice, we found that the estimates of the scaling between entrained phase and day length derived from the two approaches were in good agreement with each other (*Table 1*, *Figure 1—figure supplement 2*). Fitting details are described in the following two subsections.

Error bars in *Figure 1C*, *Figure 1—figure supplement 2A*, and *Figure 5* represent the standard deviation of peak times calculated from technical replicates (n = 4–8). Technical replicates refer to measurements obtained from side-by-side cultures subjected to the same light-dark conditions. In rare cases, cells in individual wells died or produced noisy bioluminescent signals or trajectories that fit poorly to sinusoids or parabolas. We rejected trajectories as outliers from our analysis if they produced fits with a squared error greater than 10 (0–13 outliers per 96 wells).

### (i) Estimation of period and peak times from sinusoidal regression

Sinusoidal fits were performed by least-squares minimization of the cost function:

$$\text{cost} = \sum_{i=1}^{N} \left[ y_i - \left( A \, \sin\left( \frac{2\pi x_i}{T} - \phi \right) + bx_i + C \right) \right]^2,$$

where $y_i$ and $x_i$ represent, respectively, the normalized bioluminescence signal and the time after release into constant light for the $i$-th timepoint. The period in the fits was constrained to 23 < $T$ < 25 hr. The best-fit period was distributed within these bounds, independent of the day length of the preceding entraining cycle (*Figure 1—figure supplement 2D*), a conclusion we confirmed in a peak-to-peak analysis described below.

We considered whether the quality of sinusoidal fits affected the estimated slope $m$. When we performed sinusoidal fits of the dataset in *Figure 1C*, the least-squares errors, normalized by degrees of freedom, were between 0 and 1. If data from all the wells were included in the analysis, linear regression to this data yielded a slope of $m = 0.55 \pm 0.02$ (*Table 1*). As the table below shows, imposing stricter cutoffs on least-square fitting error did not significantly impact the estimate of the slope, so an estimate of $m$ based on all of the data is reported in *Table 1*.

| Least-squares error threshold | % wells satisfying | Slope m $\pm$ SD of estimate |
|---|---|---|
| 1 | 100% | 0.55 ± 0.02 |
| 0.5 | 90% | 0.53 ± 0.01 |
| 0.1 | 45% | 0.51 ± 0.03 |
| 0.05 | 17% | 0.64 ± 0.03 |

### (ii) Estimation of period and peak times from local parabolic regression

We found that certain bioluminescent trajectories were fit poorly by sinusoids. As *Figure 1—figure supplement 3* shows, *purF* reporter waveforms display (a) strong asymmetry, marked by faster rising and slower falling dynamics (e.g. second peak in τ = 18 hr curve), (b) wide peaks (e.g. first peak in τ = 8 hr curve), and (c) broad 'shoulders' after the peak that occasionally give rise to secondary peaks (e.g. first peak in τ = 18 hr curve). In some cases, *kaiBC* reporter trajectories also exhibited successive peaks with significantly different amplitudes. In *Figure 1C* and *Figure 1—figure supplement 3*, we fit parabolas to 6 hr intervals around the peaks of normalized bioluminescence trajectories. We then estimated phases from the first peak positions and periods from the average peak-to-peak times (n = 4–8). We verified that this peak-fitting procedure produced comparable results to sinusoidal fitting for the *kaiBC* datasets in *Figure 1C* (see *Table 1* and *Figure 1—figure supplement 2B–E*).

## Comparison of waveforms during light-dark cycling and in continuous light

To compare clock reporter dynamics during entrainment and in continuous light in *Figure 1—figure supplement 1*, we computed nonparametric correlations (Kendall's τ coefficient) between the reporter signal ($P_{kaiBC}$::*luxAB*) measured during the 'day' windows in light-dark cycles (days 1–5) and during the corresponding time interval after release into continuous light (day 6). For example, in the case of LD 14:10, the normalized luminescence data recorded between 2.5 and 14 hr during each of the five entraining cycles were correlated with the luminescence dynamics measured between 2.5 and 14 hr after release into constant light. The fact that we observe Kendall's τ > 0.8 after the second driving cycle suggests that cells are effectively entrained within three cycles and that rhythms observed during the lights-on portion of a light-dark cycle can be thought of as a fragment of a free-running rhythm.

## Estimation of clock phase in vitro in metabolic cycles

For the %P-KaiC measurements in *Figure 2C*, normalized KaiC phosphorylation time courses from day 3 of each reaction were fit independently to sinusoids. Best-fit parameters were obtained by minimizing the cost function:

$$\text{cost} = \sum_{i=1}^{N}\left[ y_i - \left( A\,\sin\left(\frac{2\pi x_i}{T} - \phi\right) + C \right)\right]^2,$$

with the period *T* fixed at 24 hr to match the period of the imposed metabolic cycles.

For the fluorescence polarization experiments in *Figure 2C*, fluorescence polarization dynamics were recorded for 48 hr after the last buffer exchange, normalized (to zero mean, unit variance) and fit to sinusoids according to the expression above, with fits to all reactions from the same experiment sharing a globally fit period *T*.

## Fitting and error propagation analysis of step-response experiments

Step-response experiments described in this section were performed a total of three times: once using KaiC phosphorylation to read out clock phase and twice using the fluorescence polarization reporter of KaiB-KaiC interaction. The phosphorylation measurements were made using an independent preparation of proteins from the fluorescence polarization experiments.

We performed step-response experiments in *Figure 3* in order to determine whether the behavior of the clock in metabolic cycles could be decomposed into a sum of phase shifts due to individual transitions. To do so, we first needed (a) to extract *L* and *D* functions from step response measurements, and then (b) to use these *L* and *D* functions in numerical simulations of clock entrainment to light-dark cycles, as described in the main text and Appendix 1.

In our experiments, we directly observed how the dynamics of the fluorescence polarization reporter or KaiC phosphorylation changed as a function of lab time (e.g. *Figure 3A–B*), but for our downstream simulations and analysis (e.g. *Figure 4*) these values had to be converted to clock phase coordinates (e.g. *Figure 3C–D*) consistent with how $L(\theta)$ and $D(\theta)$ are defined. Specifically, we needed to (i) convert the lab time of each step to corresponding clock phase $\theta$ (or clock time, CT), and then (ii) determine the phase of each fluorescence polarization trace or KaiC phosphorylation trajectory after the perturbation in order to compute the phase shift ($L(\theta)$ and $D(\theta)$) relative to an unperturbed control.

Although these conversions are straightforward in principle, it is important to note that both (i) and (ii) rely on sinusoidal regression of phase from measured data, and that the best-fit phases in both cases are only estimates of the true values. The corresponding uncertainty in the best-fit parameters must be propagated through numerical simulations of entrainment.

Because the temporal resolutions of the KaiC phosphorylation and fluorescence polarization measurements are very different (3 hr vs. 15 min, respectively), we expected the sources of uncertainty in fitting these data to be different (see *Figure 2—figure supplement 1*), and we thus analyzed their errors in different ways. The sparse measurement of KaiC phosphorylation dynamics leads to relatively high uncertainty in best-fit phase and period, and we propagated the errors through our simulations using nonparametric bootstrapping of the datasets. In the case of the fluorescence polarization measurements, experiment-to-experiment variability is the major source of uncertainty. We therefore performed two replicate measurements of *L* and *D* (*Figure 3C–D*) and used the four possible combinations of *L* and *D* in entrainment simulations to assess the range of entrainment behavior constrained by our measurements of these step-response measurements. We describe each of these approaches in turn in the following two sections.

## (i) Error propagation analysis of step-response experiments performed using the fluorescence polarization reporter of KaiB-KaiC interaction

Periods and phases of reactions in each set were determined by global sinusoidal fitting: amplitude, offset and phase terms were fit independently for each reaction, but a single best-fit period was shared among all fits in a given step-up or step-down set. To avoid transient effects due to a metabolic pulse, only those data points which were collected at least 16 hr after a step-up or step-down

perturbation were used for fitting. Mathematically, we used a non-linear least-squares optimization routine to minimize the cost functions:

$$\text{cost}_{\text{day}} = \sum_{r=1}^{N_{rxns}} \sum_{i=1}^{N_r} \left[ y_{r,i} - \left( A_r \sin\left(\frac{2\pi t_{r,i}}{T_{\text{day}}} - \phi_r\right) + C_r \right) \right]^2 \text{(for reactions in day buffer)}$$

and

$$\text{cost}_{\text{night}} = \sum_{r=1}^{N_{rxns}} \sum_{i=1}^{N_r} \left[ y_{r,i} - \left( A_r \sin\left(\frac{2\pi t_{r,i}}{T_{\text{night}}} - \phi_r\right) + C_r \right) \right]^2 \text{(for reactions in night buffer)},$$

where $y_{r,i}$ refers to the $i$-th data point in the $r$-th normalized fluorescence polarization trajectory, $t_{r,i}$ refers to the lab time when that data point was collected, and $N_r$ is the number of data points fit in the $r$-th reaction. Here $A_r$, $\phi_r$ and $C_r$ are the best-fit parameters for the $r$-th reaction; $T_{day}$ and $T_{night}$ are best-fit periods for all reactions in the 100% ATP and 25% ATP datasets, respectively.

These best-fit parameters were used to determine the phase $\theta$ of each step perturbation and corresponding phase shifts $L(\theta)$ and $D(\theta)$, analogously to the way described for the KaiC phosphorylation datasets below. The step-response functions were interpolated linearly to generate smooth curves while enforcing $2\pi$-periodicity.

Linearized step-response functions $L_{lin}(\theta)$ and $D_{lin}(\theta)$ were prepared by linear regression of $L(\theta)$ and $D(\theta)$ centered on the regions used during metabolic entrainment: $D$ functions were linearized between 6 and 22 CT hr; $L$ functions were linearized between 18 and 34 CT hr (see *Figure 3C–D*). To satisfy periodicity, the step-response functions were assembled as piecewise-linear functions with the same slope everywhere but with offsets every $2\pi$ radians at 'breakpoints,' which were selected by manual inspection. See *Figure 4—figure supplement 2C* for an illustration of $L(\theta)$ and $D(\theta)$ and the corresponding $L_{lin}(\theta)$ and $D_{lin}(\theta)$.

Finally, step phases deduced from polarization trajectories were adjusted to match the phases of the KaiC phosphorylation rhythm. This conversion made use of measurements in *Figure 2—figure supplement 1*, which indicated that the phase of oscillation in KaiC phosphorylation lags behind the phase of the polarization reporter of KaiB-KaiC binding by approximately $2\pi/3$.

By performing this analysis for the two step-up and two step-down datasets in *Figure 3C–D*, we generated two sets of $\{L, L_{lin}, T_{day}, T_{night}\}$ and two sets of $\{D, D_{lin}, T_{day}, T_{night}\}$. The phase oscillator simulations described below require using the $L$ and $D$ functions together in each simulation. To propagate the experiment-to-experiment variability in step-response measurements through our simulations, we combined the two measurements of $L$ and two measurements of $D$ into the four possible combinations of $\{L, D\}$ pairs. This resulted in four sets of $\{L, L_{lin}, D, D_{lin}, T_{day}, T_{night}\}$, which were used in the simulations below.

The ratio of oscillator frequencies in dark and light, $\frac{\omega_D}{\omega_L} = \frac{T_{day}}{T_{night}} = 0.93 \pm 0.01$, was determined by averaging the values of $T_{\text{day}}$ and $T_{\text{night}}$ from the two step-up and two step-down sets above. Best estimates for the slopes $l$ and $d$ in *Figure 4—figure supplement 3* were computed via the following bootstrap analysis. For each $L$ and $D$ function in *Figure 3C–D*, we selected points from the regions used for linearization above (6–22 CT hr on $D$, 18–34 CT hr on $L$). We sampled these data points with replacement until we generated 500 samples containing at least three unique points. For each set of resampled points, we computed the slope of the best-fit line, for a total of 1000 samples of $l$ and $d$. The crosshair in *Figure 4—figure supplement 3* marks the average $\pm$ standard deviation of these values ($l = 0.34 \pm 0.03$, $d = 0.38 \pm 0.05$).

The phase oscillator model discussed in the main text and the Appendix 1 makes the prediction that the proportionality constant $m$ between entrained phase and day length depends on $l$ and $d$ coefficients according to:

$$m(l,d) = 1 - \frac{-(1-l) \times (\omega_D/\omega_L)}{d+l-ld} - \frac{1}{d+l-ld}.$$

To check whether this prediction is in line with our experimental measurement of $m$, we used this formula to calculate $m$ for every pair of $l$ and $d$ samples generated in the bootstrapping procedure above, assuming $\omega_D/\omega_L = 0.93$. According to this calculation, $m(l,d) = 0.34 \pm 0.04$ (mean $\pm$ standard

deviation of the distribution), which is in agreement with the experimental measurement in **Figure 2C** ($m = 0.38 \pm 0.07$).

## (ii) Non-parametric bootstrapping of step-response datasets collected using SDS-PAGE analysis of KaiC phosphorylation

Recall that measuring each step-response function requires (a) conversion of the lab time of each step to corresponding clock phase $\theta$ (or clock time, CT), and then (b) determination of the phase of each KaiC phosphorylation trajectory after the perturbation in order to compute the phase shift ($L(\theta)$ and $D(\theta)$) relative to an unperturbed control. In particular, uncertainties in (a) manifest as phase errors on $L$ and $D$ in **Figure 3E–F**; these errors are correlated across all points on the step-response function. Uncertainties in (b) manifest as phase shift errors on $L$ and $D$; these errors derive from the errors in the phase estimates of both the control and step reactions. To propagate both these sources of error, we used the following non-parametric bootstrapping strategy.

First, KaiC phosphorylation dynamics from all step-response measurements were assembled into a master dataset containing nine step-up trajectories, nine step-down trajectories, as well as two control reactions. These measurements were performed once on an independent preparation of proteins from the batch used to generate data in **Figure 3C–D**. This master dataset was then trimmed to include only data collected at least 16 hr after a step transition, and every trajectory was normalized. To generate bootstrapped datasets, we sampled with replacement from the entire master dataset (as opposed to resampling reactions individually) 1000 times.

Next, we used each resampled dataset to compute phase shifts in KaiC phosphorylation due to step-up and step-down transitions; in other words, each bootstrapped dataset was used to derive a bootstrapped pair of $L(\theta)$ and $D(\theta)$. For a given dataset, we globally fit all 100% ATP trajectories (nine step-up trajectories, plus the 100% ATP control reaction) such that all fits shared a best-fit period ($T_{day}$), but phase, amplitude and offset terms were fit independently for each reaction, as described above for polarization datasets. Likewise, we fit all 20% ATP trajectories (nine step-down reactions, plus the 20% ATP control reaction) to obtain a globally best-fit period ($T_{night}$) and independently fit phases for every reaction.

$L(\theta)$ and $D(\theta)$ map the phase at which a metabolic step occurs to the resulting phase shift. The phase at which the metabolic step occurred for every reaction $r$ was computed from the best-fit phase of the appropriate control reaction at step time $t_{r,step}$:

$$\theta^{control}_{r,step} = \frac{2\pi t_{r,step}}{T_{con}} - \phi_{con},$$

where $\phi_{con}$ is the best-fit phase of the control reaction and $T_{con}$ is the globally-fit oscillator period in the appropriate control condition (i.e. $T_{con} = T_{day}$ for step-down reactions, $T_{con} = T_{night}$ for step-up reactions).

Similarly, we computed the apparent phase of each perturbed reaction at the time of each step:

$$\theta_{r,step} = \frac{2\pi t_{r,step}}{T_{pert}} - \phi_r,$$

where $\phi_r$ is the best-fit phase of the $r$-th reaction and $T_{pert}$ is the globally-fit oscillator period in the perturbed condition ($T_{pert} = T_{night}$ for step-down reactions, $T_{pert} = T_{day}$ for step-up reactions).

Finally, we defined the phase shift in response to each step perturbation as the difference in phase of the perturbed reaction and the appropriate unperturbed control.

$$L\left(\theta^{control}_{r,step-up}\right) = \theta_{r,step-up} - \theta^{control}_{r,step-up}.$$

$$D\left(\theta^{control}_{r,step-down}\right) = \theta_{r,step-down} - \theta^{control}_{r,step-down}.$$

To estimate phase shifts at other values of $\theta$, we linearly interpolated $L$ and $D$ between the measured values while enforcing $2\pi$-periodicity.

For each set of $L$ and $D$ generated in this way, we also prepared linearized versions $L_{lin}$ and $D_{lin}$. Linear fits to $L$ and $D$ were performed over the range of step times similar to the ones we selected

for fluorescence polarization-based step-response functions as discussed above ($L$ between 18 and 33 CT hr, $D$ between 6 and 23 CT hr). These regions of $L$ and $D$ were selected by visual inspection; they are centered on phases used by the KaiABC oscillator in diurnal cycles (i.e. see arrows in *Figure 3E–F*) but also contain enough points (5-7) to avoid biasing the slope estimate by a single poorly fit data point.

We extrapolated the linear approximations to $L$ and $D$ over the entire cycle, with a single breakpoint away from the linear region to satisfy $2\pi$-periodicity. Breakpoints were selected by visual inspection. While interpolating between data points near the breakpoint, we assumed that $L$ and $D$ (or $L_{lin}$ and $D_{lin}$) never generate phase shifts larger than one cycle (i.e. winding number of 0). We anticipate that this choice does not significantly affect entrained phase in most of our simulations because the regions of $L$ and $D$ near the breakpoint are rarely used by the oscillator during entrainment when $\tau < 14$ hr.

We repeated this procedure for each of the 1000 bootstrapped datasets, thereby generating 1000 sets of $\{L, D, L_{lin}$ and $D_{lin}, T_{day}, T_{night}\}$ that were used for subsequent analysis. *Figure 3E–F* shows the mean ± standard deviation of the distribution of bootstrapped $L$ and $D$ generated this way.

## Simulations of a phase oscillator driven by a light-dark cycle

We simulated entrainment to a step-like driving cycle for a phase oscillator governed by the four combinations of $\{L, L_{lin}, D, D_{lin}, T_{day}, T_{night}\}$ determined from the fluorescence polarization assays and the 1000 bootstrapped sets of $\{L, D, L_{lin}, D_{lin}, T_{day}, T_{night}\}$ derived from KaiC phosphorylation measurements. Electing to work in units of cycles (1 cycle = $2\pi$ rad), we defined a phase variable $\hat{\theta} = (\theta - \pi/2)/(2\pi)$, such that $\hat{\theta} = 0$ cycles and $\hat{\theta} = 1$ cycles correspond to the trough of the KaiC phosphorylation trajectory and $\hat{\theta} = 0.5$ cycles corresponds to the peak. Where circadian time (CT) is mentioned in the text, we have adopted the convention that CT = 0 and 24 hr refer to $\hat{\theta} = 0$ and 1 cycles, respectively; CT = 12 hr refers to $\hat{\theta} = 0.5$ cycles. We also defined corresponding analogs of $L$ and $D$ in units of cycles:

$$\hat{L}\left(\hat{\theta}\right) = \frac{L(\theta)}{2\pi}$$
$$\hat{D}\left(\hat{\theta}\right) = \frac{D(\theta)}{2\pi}$$

See *Figure 4—figure supplement 2C* for examples of $\hat{L}, \hat{D}, \hat{L}_{lin}$ and $\hat{D}_{lin}$. For each set of $\left\{\hat{L}, \hat{D}, \hat{L}_{lin}, \hat{D}_{lin}, T_{day}, T_{night}\right\}$ generated in this way, we modeled oscillator entrainment to a driving cycle. As shown schematically in *Figure 4A* and the Appendix 1, oscillator phase increases with constant angular velocity (given by $\omega_L = \frac{1}{T_{day}}$ during the day and $\omega_D = \frac{1}{T_{night}}$ during the night); at dawn and dusk, oscillator phase shifts instantaneously according to $\hat{L}$ and $\hat{D}$, respectively. Therefore, oscillator phases at dusk and dawn of the $n$-th entraining cycle can be computed iteratively:

$$\hat{\theta}_n^{dawn} = \hat{\theta}_{n-1}^{dusk} + \hat{D}\left(mod\left(\hat{\theta}_{n-1}^{dusk}, 1\right)\right) + \frac{T - \tau}{T_{night}}$$

$$\hat{\theta}_n^{dusk} = \hat{\theta}_n^{dawn} + \hat{L}\left(mod\left(\hat{\theta}_n^{dawn}, 1\right)\right) + \frac{\tau}{T_{day}}$$

We simulated oscillator behavior in 24 hr light-dark cycles (driving period $T_{dr}$ = 24) with day length $\tau$ lasting from 4 to 18 hr. In simulations relying on $\hat{L}$ and $\hat{D}$ determined from KaiC phosphorylation measurements, we set the initial condition $\hat{\theta}_0$ for a given value of $\tau$ based on our measurements of entrained phase of the KaiABC oscillator in corresponding metabolic cycles (i.e. based on values interpolated between black markers in *Figure 2C*). Simulations using step-response functions based on fluorescence polarization measurements were started from $\hat{\theta}_0 = 0$. We simulated 10 light-dark cycles and recorded oscillator phase after each dawn (immediately after $\hat{L}\left(\hat{\theta}\right)$ phase shift).

For each set of $\hat{L}$ and $\hat{D}$ we performed simulations for $\tau$ = 4–18 hr for both the phosphorylation-based step functions and fluorescence polarization-based step functions. *Figure 4—figure*

supplement 1 illustrates simulations governed by one such set of $\hat{L}$ and $\hat{D}$ (derived from the magenta and blue step-response measurements shown in *Figure 3C–D*). We also carried out entrainment simulations for the linearized step-response functions described above (*Figure 4—figure supplement 2*).

We judged that the oscillator entrained stably to a given diurnal cycle if the standard deviation of clock phases at the dawns of the last five driving cycles was less than 0.01 cycles. For each value of τ, we selected those simulations where the oscillator entrained stably to the light-dark cycle, and computed the mean phase at dawn and its standard deviation (σ) for that set of simulations. These values are represented in shaded areas in *Figure 4C* , *Figure 4—figure supplement 2A* (for fluorescence polarization data) and *Figure 4—figure supplement 2B* (for KaiC phosphorylation data).

We were also interested in how quickly the oscillator approached entrainment in simulations for 6 ≤ τ ≤18, the range we profiled experimentally in *Figure 2*. To make this determination for a given value of τ, we computed how much oscillator phase at dawn varied over three successive cycles in a sliding window:

$$EC_n = \sqrt{var\left( \hat{\theta}_n^{dawn}, \hat{\theta}_{(n+1)}^{dawn}, \hat{\theta}_{(n+2)}^{dawn} \right)} \text{ for n} = 1 - 8$$

We used the first value of *n* for which $EC_n < 0.01$ as a proxy for the speed of approach to entrainment.

The tables below display summary statistics for the entrainment simulations.

## Simulations based on step-response functions measured via fluorescence polarization reporter

| Step fun. type | τ's profiled | No. sim. | % entrained sim. | | % entrained within three cycles (8 < τ < 16) | |
|---|---|---|---|---|---|---|
| | | | 4 < τ < 18 | 8 < τ < 16 | all sim. | all sim. entrained within eight cycles |
| $\hat{L}$ and $\hat{D}$ | 4, 4.01, …, 18 hr | 564 | 100% | 100% | 83% | 83% |
| $\hat{L}_{lin}$ and $\hat{D}_{lin}$ | 4, 4.01, …, 18 hr | 564 | 100% | 100% | 100% | 100% |

## Simulations based on step-response functions measured via SDS-PAGE analysis of KaiC phosphorylation

| Step fun. type | τ's profiled | No. sim. | % entrained sim. | | % entrained within three cycles (8 < τ < 16) | |
|---|---|---|---|---|---|---|
| | | | 4 < τ < 18 | 8 < τ < 16 | all sim. | all sim. entrained within eight cycles |
| $\hat{L}$ and $\hat{D}$ | 4, 4.25, …, 18 hr | 57 000 | 76% | 87% | 86% | 97% |
| $\hat{L}_{lin}$ and $\hat{D}_{lin}$ | 4, 4.25, …, 18 hr | 57 000 | 98% | 100% | 99% | 100% |

In the large majority of our simulations, we found that the phase oscillator entrained stably within three light-dark cycles. For simulations derived from KaiC phosphorylation datasets, we found that the oscillator either entrained to the driving cycle quickly or not at all. Indeed, when we restricted our analysis only to those simulations that were judged as entrained within eight cycles, over 96% entrained within three light-dark cycles. Generally, the oscillator entrained readily for day lengths shorter than 14 hr, but often failed to entrain for longer day lengths. We determined that this occurs because for τ >14 dawn phases often sample the *L* function near the breakpoint of the curve (near 18 CT hr in *Figure 3F*), leading to erratic responses to driving cues (i.e. lack of entrainment) or disagreement between simulations and experiment (*Figure 4—figure supplement 2B*). Relatedly, we

believe that the better agreement with experiment achieved in simulations using the step functions derived from the fluorescence polarization data than from the KaiC phosphorylation data reflects the better temporal resolution of the breakpoint (2 hr using the polarization approach vs. 4 hr for KaiC phosphorylation).

In simulations of entrainment to light-dark cycles of varying period in *Figure 3—figure supplement 2*, we relied on a single set of $\{L, D, L_{lin}, D_{lin}, T_{day}, T_{night}\}$ based on the step-response measurements shown in blue and magenta in *Figure 3C–D*. We simulated entrainment to driving periods from 6 to 48 hr, in increments of 0.0025 hr. For each driving period $T_{drive}$, we subjected the phase oscillator to 1000 cycles with equal day and night durations ($\tau - T_{drive}/2$) and plotted the phases attained by the oscillator at the end of nighttime (immediately preceding the action of $L$) at the last 950 cycles.

## Simulations of phase resetting

We used $\hat{L}_{lin}$ and $\hat{D}_{lin}$ derived from the same step-response measurements as in *Figure 4—figure supplement 1* to simulate response of a phase oscillator to 12 hr dark pulses administered throughout the circadian cycle (*Figure 5—figure supplement 1*). Phase evolution of the oscillator was simulated explicitly for 120 hr using a timestep of $dt = 0.01$ hr:

$$\hat{\theta}_{t+dt} = \begin{cases} \hat{\theta}_t + \frac{dt}{T_{day}} & \text{(in light)} \\ \hat{\theta}_t + \frac{dt}{T_{night}} & \text{(in dark)} \\ \hat{\theta}_t + \hat{L}\left(mod\left(\hat{\theta}_t, 1\right)\right) & \text{(at dawn)} \\ \hat{\theta}_t + \hat{D}\left(mod\left(\hat{\theta}_t, 1\right)\right) & \text{(at dusk)} \end{cases}$$

## Estimation of phase shifts in response to dark pulses in vivo

For phase resetting and wedge experiments (*Figure 5*), clock phase was estimated by sinusoidal regression of normalized bioluminescence data collected 36–48 hr after the end of the applied dark pulse. Phase shifts in response to dark pulses were computed as differences in average peak times between perturbed wells ($t_{pk,DP}$) and controls ($t_{pk,LL}$): $\Delta t_{pk} = \bar{t}_{pk,DP} - \bar{t}_{pk,LL}$, where overbars indicate averages over replicate wells (n = 4–8). Clock phases at which the dark pulses were applied were determined from the average fit phase and period of the unperturbed (control) wells ($\bar{\phi}_{LL}$ and $\bar{T}_{LL}$) according to $\hat{\theta}_t = (t/\bar{T}_{LL}) - (\bar{\phi}_{LL} + 0.5\pi)/(2\pi)$, where $\hat{\theta}_t$ is measured in cycles and $\hat{\theta} = 0$ corresponds to the minimum of an oscillatory trajectory.

## Global fit to phase response and seasonal entrainment datasets

In the Appendix 1, we show that in the regime where the circadian clock is well-approximated by a phase oscillator governed by linear $L$ and $D$ step-response functions, the slopes of seasonal entrainment and phase resetting of the clock can be described by a model with two free parameters $\beta_1$ and $\beta_2$: for phase resetting, $\Delta t_{pk} = \hat{\theta}_t/(\omega_L \beta_2) + \delta(1 + \beta_1/\beta_2) + C_1$ and for entrainment $t_{pk} = \tau(1 - \beta_1 - \beta_2) + T\beta_1 + C_2$. Here, $\delta$ is dark pulse duration (in hours), $\tau$ is day length (in hours), $\hat{\theta}_t$ is the clock phase at time $t$ (in cycles), $\omega_L$ is the clock frequency in the light (in units of cycles/hour), $T$ is the driving period (in hours), and $C_1$ and $C_2$ are constants that do not depend on $\hat{\theta}_t$, $\delta$, $\tau$, or $T$. For phase resetting experiments, $\hat{\theta}_t$ and $\omega_L$ were estimated based on the average of the best sinusoidal fits to unperturbed wells in each experiment ($\bar{\phi}_{LL}$ and $\bar{T}_{LL}$). In particular, we set $\hat{\theta}_t = \omega_L t - (\bar{\phi}_{LL} + 0.5\pi)/(2\pi)$. In the formula for $\Delta t_{pk}$, the term $\hat{\theta}_t/(\omega_L \beta_2)$ thus simplifies to $(t/\beta_2) - (\bar{\phi}_{LL} + 0.5\pi)/(2\pi\beta_2)$, and the term to the right of the minus sign was incorporated into the constant term $C_1$.

In the global fits of all datasets in *Figure 5A–C*, we varied $\beta_1$ and $\beta_2$ to simultaneously fit $\Delta t_{pk}$ to our phase-resetting and wedge data and $t_{pk}$ to our seasonal entrainment data. The constant terms were allowed to vary as follows:

- in *Figure 5C*, a single $C_2$ term was fit for all curves, referred to as $C_{entrainment}$ below;
- in *Figure 5B*, a single $C_1$ intercept was fit for both datasets ($t_{DP} = 36$ and $t_{DP} = 39$ hr), referred to as $C_{wedge}$ below;

- in *Figure 5A*, the points before and after the breakpoint were fit using the same slope, but varying constant ($C_1$) terms. The breakpoint was selected by visual inspection. Below, the intercepts to the left and right of the breakpoint are referred to as $C_{PRC\_left}$ and $C_{PRC\_right}$, respectively.

In total, only two parameters ($\beta_1$ and $\beta_2$)) were used to determine the slopes of all curves, and six parameters were used for the entire global fit of nine linear segments, which minimized the cost function:

$$\chi^2 = \sum_{i=1}^{N} \left( \frac{\hat{y}_i - y_{fit,i}}{\sigma_i} \right)^2,$$

where each $\hat{y}_i$ represents the average measurement of phase shift or peak time and $\sigma_i$ represents the standard error of that measurement. The reduced chi-squared value of the fit was $\chi_\nu^2 = 5.67$. The best-fit coefficients determined in the fit are summarized in the table below.

| Parameter | Best-fit value | 95% CI |
|---|---|---|
| $\beta_1$ | −1.31 | [−1.46 −1.17] |
| $\beta_2$ | 1.79 | [1.64 1.95] |
| $C_{PRC\_left}$ | −22.11 | [−24.3 −19.9] |
| $C_{PRC\_right}$ | −36.98 | [−40.0 −34.0] |
| $C_{wedge}$ | −24.52 | [−26.6 −22.5] |
| $C_{entrainment}$ | 39.8 | [36.3 43.3] |

## Simulations of entrainment in the limit cycle geometry model

To simulate entrainment to light-dark cycles of different day length in the geometric resetting framework in *Figure 6*, we modeled an oscillator running along the daytime limit cycle (centered at 0, radius 1) in the light and the nighttime cycle (centered at $X$, radius $R$) in the dark. We set the angular frequency to be $\frac{2\pi \text{ rad}}{24 \text{ hr}}$ in both light and dark. As described in the Appendix 1, dusk and dawn transitions were modeled as radial jumps from one cycle to the nearest point on the other cycle.

We considered values of $R$ and $X$ spanning six orders of magnitude and studied entrainment to 24 hr cycles with day length lasting from 6 to 18 hr. For each pair of $R$ and $X$, we simulated oscillator dynamics in 30 light-dark cycles of a given day length. We judged that an oscillator failed to entrain to a given diurnal schedule if oscillator phases after 29 and 30 cycles were more than $\pi/180$ radians apart, or if simulations starting from different initial phases ($\theta_0 = \pi/4$ and $5\pi/4$) reached phases over $\pi/180$ radians apart after 30 light-dark cycles.

For every simulation that passed the entrainment criteria above, we computed the best linear fit and slope $m$ of oscillator phase dependence on day length $\tau$. We assessed goodness of fit by computing the mean fit error, defined as the average absolute value of the deviations between the linear fit and simulation results. If the mean fit error was greater than 10% of the deviation between maximum and minimum phases to which the oscillator entrained for this range of $\tau$, the phase dependence on day length was judged to be non-linear.

## Relaxing assumptions of the limit cycle geometry model

The results of the simulations described immediately above strongly suggest that the relative geometry of day and night orbits determines the scaling of entrained phase with day length. However, those simulations are based on idealized infinitely-attracting circular limit cycles with constant angular frequency, assumptions which are likely to be violated for biological clocks. For example, the KaiC phosphorylation limit cycles in *Figure 6F* are somewhat elliptical. We were therefore interested in understanding whether the relative geometry of the limit cycles would be the dominant determinant of oscillator entrainment if our assumptions were relaxed. To this end, we explored entrainment in limit cycle models where these features—(i) orbit attraction strength, (ii) orbit ellipticity, and (iii) variation in angular frequency throughout the cycle—could be treated explicitly.

### (i) Orbit attraction strength

To consider the effect of orbit attraction strength (*Figure 6—figure supplement 2A*), we studied an oscillator with constant angular frequency $\omega$ orbiting a circular limit cycle of radius $R_{orb}$ that is exponentially attracting:

$$\dot{\theta} = \omega = \frac{2\pi}{24}\frac{\text{rad}}{\text{hr}}$$
$$\dot{r} = -a(r - R_{orb}),$$

where the polar coordinates ($r$, $\theta$) are defined relative to the center of the limit cycle. Here, the attraction strength $a$ determines the half-time for relaxation to the orbit according to $t_{1/2} = \ln(2)/a$ (hr). During the day, the equations of motion were integrated with respect to the daytime orbit of radius 1 centered at the origin. At night, the oscillator coordinates were computed with respect to the night limit cycle of radius $R$ centered at (0, $X$). We considered geometries with $R$ and $X$ ranging over six orders of magnitude and $a$ = 10, 1, and 0.1. For each set of $R$, $X$ and $a$, we simulated entrainment to ten 24 hr light-dark cycles of day length $\tau$ between 6 and 18 hr starting from two out-of-phase initial conditions ($\theta_0$ = 0 and $\theta_0$ = $\pi$). After the end of the tenth entraining cycle, the oscillator was allowed to relax back to the daytime orbit. We then computed the 'peak time' ($t_{pk}$) of this oscillator as the additional time required to reach phase $\theta$ = $\pi/2$ after return to the orbit. For every simulation that passed the entrainment criteria defined above, we computed the slope $m$ of the best linear fit of oscillator peak time dependence on day length $\tau$. We assessed goodness of fit by computing the mean fit error, defined as the average absolute value of the deviations between the linear fit and simulation results. The phase dependence on day length was judged to be non-linear if the mean fit error was greater than 0.5 hr and also greater than 10% of the deviation between maximum and minimum peak times to which the oscillator entrained for this range of $\tau$.

### (ii) Orbit ellipticity

To consider the effect of orbit ellipticity (*Figure 6—figure supplement 2B*), we studied an oscillator with constant angular frequency $\omega$ orbiting an elliptical limit cycle that is exponentially attracting. We only considered orbits with their minor axes (length $R_{orb}$) positioned along the x axis and major axes (length $\rho \times R_{orb}$) lying parallel to the y axis. Mathematically, such an oscillator is defined by:

$$\dot{\theta} = \omega = \frac{2\pi}{24}\frac{\text{rad}}{\text{hr}}$$
$$\dot{r} = -a \times d_{ellipse}(r, \theta) - \dot{r}_{ellipse}(\theta),$$

where the polar coordinates ($r$, $\theta$) are defined relative to the center of the limit cycle. Here $d_{ellipse}(r, \theta)$ is the distance from the current point to the nearest point on the ellipse, and $\dot{r}_{ellipse}(\theta)$ measures how the radial coordinate changes as a function of the angle on an elliptical trajectory, assuming that $\dot{\theta}$ is constant. $d_{ellipse}(r, \theta)$ was evaluated numerically at every integration time step using the fminbnd() routine in MATLAB. $\dot{r}_{ellipse}(\theta)$ was computed explicitly based on the definition of the ellipse in polar coordinates:

$$\dot{r}_{ellipse}(\theta) = -\frac{1}{2}\left(\frac{\cos^2\theta}{R_{orb}^2} + \frac{\sin^2\theta}{\rho^2 R_{orb}^2}\right)^{-3/2}\left(\frac{-2\dot{\theta}\cos\theta\sin\theta}{R_{orb}^2} + \frac{2\dot{\theta}\cos\theta\sin\theta}{\rho^2 R_{orb}^2}\right).$$

For all simulations in *Figure 6—figure supplement 2B*, the day orbit was centered at (0, 0) with minor axis length 1 and major axis length $\rho_D$; the night orbit was centered at (0, $X$) with minor axis length $R$ and major axis length $\rho_N R$. The attraction strength of both orbits was set to $a$ = 1. Entrainment simulations were carried out for four light-dark cycles. Slope $m$ was computed as in (i) above.

### (iii) Varying angular velocity along the limit cycle

We also considered the case of circular orbits with nonconstant angular velocities (*Figure 6—figure supplement 2C*). To do so, we considered an oscillator defined by the following equations:

$$\dot{\theta} = \omega\left(1 + \frac{\varepsilon}{\omega}\sin\omega t\right)$$
$$\dot{r} = -a(r - R_{orb}),$$

where $\omega = \frac{2\pi}{24}\frac{\text{rad}}{\text{hr}}$ is the natural oscillator frequency. As above, the polar coordinates ($r$, $\theta$) are defined

relative to the center of the limit cycle. Such an oscillator completes one full cycle around the orbit within 24 hr, but the oscillator speed varies sinusoidally along the orbit. Here, the ratio $\varepsilon/\omega$ defines the maximal deviation of the angular velocity from the natural frequency. In the most perturbative case we considered, $\frac{\varepsilon}{\omega} = \frac{1}{2}$, the oscillator runs at 1.5 times the natural frequency at the peak of the oscillation and at 0.5 times natural frequency at the trough of the cycle. Simulations were carried out for ten light-dark cycles for orbits with attraction strength $a = 1$. Slope $m$ was computed as in (i) above.

## Simulations of daytime resource allocation in days of different length

Phase of transcriptional dynamics of dawn and dusk genes in **Figure 6—figure supplement 3** were derived from a sinusoidal fit to the transcriptional profiles of dawn and dusk genes identified in microarray time courses by **Vijayan et al. (2009)**. Briefly, normalized time course data was downloaded from the GEO depository, and time series for transcripts annotated as dawn genes were averaged to obtain the average dawn gene transcriptional profile. The average dusk gene trajectory was obtained analogously. Average dawn and dusk waveforms were fit to sinusoids with 24 hr period, and the resulting phase estimates were used to define the phase shift between the orange and gray curves in **Figure 6—figure supplement 3**.

In **Figure 6—figure supplement 3**, nightfall in LD 12:12 for all values of $m$ coincides with the 12 hr dark pulses administered during the initial synchronization in the Vijayan et al. experiment. In simulations of other day lengths ($\tau$), we assumed that the only effect of diurnal cycling is to adjust the phase of the circadian transcriptional program relative to dawn and dusk, without affecting the shape or relative timing of dawn and dusk gene transcriptional waveforms. The bias in allocation of daytime resources between dawn and dusk genes was then computed according to the expression:

$$allocation\ bias = \frac{I(dawn\ gene) - I(dusk\ gene)}{I(dawn\ gene) + I(dusk\ gene)},$$

where $I(gene)$ is the integrated RNA signal for that gene over the daytime hours.

## Acknowledgements

We thank Arvind Murugan and members of the Rust and Dinner labs, particularly H Gudjonson and G Pattanayak, for feedback and useful discussions. Joel Heisler and Andy LiWang assisted us with fluorescence polarization measurements. This work was supported by NIH Training Grant 5T32EB009412 (EL), NIH Training Grant T32GM07197 (UL), NIH R01 GM107369-01 (to MJR), NSF PHY-1305542 (to ARD), a Pew Biomedical Scholarship (to MJR). EL, JL, HY, and UL carried out experiments. EL analyzed data. EL, ARD and MJR designed the study, performed mathematical modeling, and wrote the paper.

## Additional information

### Funding

| Funder | Grant reference number | Author |
| --- | --- | --- |
| National Institutes of Health | 5T32EB9412-5 | Eugene Leypunskiy |
| National Institutes of Health | T32GM07197 | UnJin Lee |
| National Science Foundation | PHY-1305542 | Aaron R Dinner |
| Pew Charitable Trusts | | Michael J Rust |
| National Institute of General Medical Sciences | R01GM107369-01 | Michael J Rust |

The funders had no role in study design, data collection and interpretation, or the decision to submit the work for publication.

## Author contributions
EL, Conceptualization, Software, Investigation, Methodology, Writing—original draft, Writing—review and editing; JL, UJL, Investigation; HY, Methodology; ARD, Conceptualization, Formal analysis, Funding acquisition, Writing—original draft, Project administration, Writing—review and editing; MJR, Conceptualization, Funding acquisition, Investigation, Writing—original draft, Project administration, Writing—review and editing

## Author ORCIDs
Eugene Leypunskiy, http://orcid.org/0000-0002-3335-1099
Aaron R Dinner, http://orcid.org/0000-0001-8328-6427
Michael J Rust, http://orcid.org/0000-0002-7207-4020

## Additional files

### Major datasets
The following previously published dataset was used:

| Author(s) | Year | Dataset title | Dataset URL | Database, license, and accessibility information |
|---|---|---|---|---|
| Vijayan V, Zuzow R, O'Shea EK | 2009 | S. elongatus circadian microarray | https://www.ncbi.nlm.nih.gov/geo/query/acc.cgi?acc=GSE18902 | Publicly available at the NCBI Gene Expression Omnibus (accession no: GSE18902) |

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

## Appendix 1

### Mathematical

Here we outline a simple and general mathematical framework for describing the interaction between a circadian clock and its environment. The essential idea is that we model only the phase of the oscillator and decompose its dynamics into a sequence of free runs interrupted by step responses to changes in the environment (*Appendix 1—figure 1*). We present the assumptions of the model and then derive key formulas for interpreting our measurements.

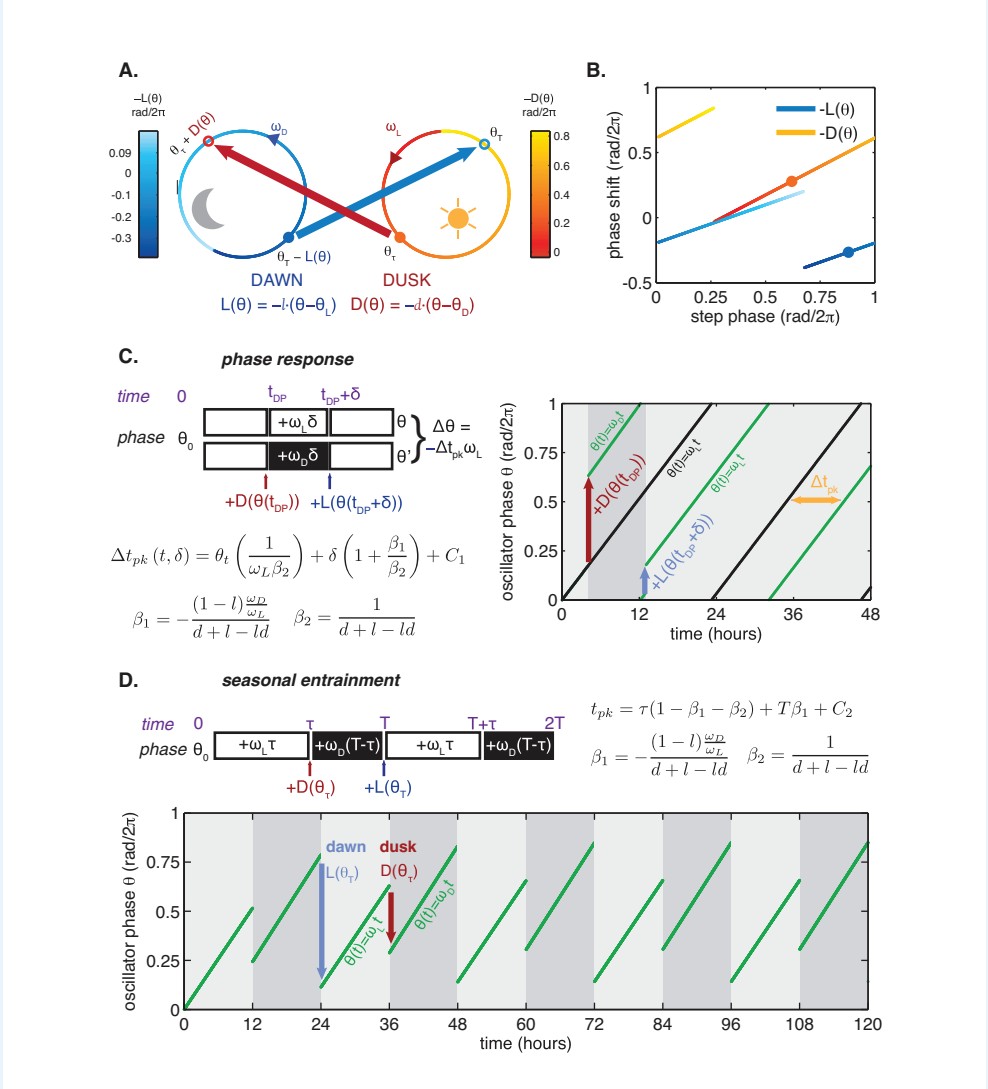

**Appendix 1—figure 1.** Phase oscillator with step response framework. (**A**) Schematic of the framework. In the light portion of the day, the oscillator runs along the 'light' limit cycle (*orange*) and accumulates phase $\theta$ at constant rate $\omega_L$; in the dark, the oscillator runs along the 'dark' limit cycle (*blue*) with frequency $\omega_D$. At dawn (*solid blue circle*) and dusk (*solid orange circle*), the oscillator transitions between the light and dark limit cycles (*blue and orange arrows*) and incurs instantaneous phase shifts given by $L(\theta)$ at dawn and $D(\theta)$ at dusk. $L(\theta)$ and $D(\theta)$ can be approximated by linear functions of phase, such that $L(\theta) = -l(\theta - \theta_L)$ and $D(\theta) = -d(\theta - \theta_D)$. Colors along the 'light' and 'dark' limit cycles indicate the magnitude of phase shifts due to light-to-dark ($D(\theta)$) and dark-to-light ($L(\theta)$) responses

throughout the cycle for a single set of linearized $L(\theta)$ and $D(\theta)$, shown in (**B**), generated by bootstrapping the measurements in ***Figure 3E–F***. $\theta = 0$ refers to the minimum of the KaiC phosphorylation rhythm. (**B**) One set of linearized step response functions $\hat{L}_{lin}\left(\hat{\theta}\right)$ and $\hat{D}_{lin}\left(\hat{\theta}\right)$ for the KaiABC oscillator determined by SDS-PAGE analysis of KaiC phosphorylation rhythms (see ***Figure 3E–F***). The functions plotted represent one pair of $\hat{L}_{lin}\left(\hat{\theta}\right)$ and $\hat{D}_{lin}\left(\hat{\theta}\right)$ generated by bootstrapping (see Computational Methods). Colors denote phase shift magnitude as in (**A**). Solid circular markers correspond to dawn and dusk transitions in (**A**). (**C**) (*left*) Schematic of phase response analysis in the phase oscillator framework. The phase shift is a linear function of the dark pulse time ($t_{DP}$) and duration ($\delta$) if $L(\theta)$ and the $D(\theta)$ are linear functions. Slopes of linear dependencies on $t_{DP}$ and duration $\delta$ can be computed based on clock frequencies in light and dark ($\omega_L$, $\omega_D$) and slopes of $L(\theta)$ and $D(\theta)$ functions. (*right*) Simulation of a phase response experiment for a phase oscillator with experimentally determined parameters ($\omega_L$, $\omega_D$, $l$, and $d$ from the same bootstrapped parameter set as in (**B**)). A 9 hr dark pulse applied 4 hr after the beginning of the simulation to an oscillator (*green*) results in a ≈8 hr phase delay relative to a control (*black*) that remains in light throughout the simulation. (**D**) (*top*) Schematic of analysis of seasonal entrainment in the phase oscillator framework. Entrained phase is a linear function of day length ($\tau$) and driving period (T) if $L(\theta)$ and the $D(\theta)$ are linear functions. Slopes of linear dependencies on $\tau$ and $T$ can be computed based on four parameters ($\omega_L$, $\omega_D$, $l$, and $d$). (*bottom*) Simulation of entrainment to a LD 12:12 cycle for a phase oscillator with experimentally determined parameters ($\omega_L$, $\omega_D$, $l$, and $d$ from the same bootstrapped parameter set as in (**B**)).

## Modeling a circadian clock as a phase oscillator with step responses to light-dark cues

Circadian rhythms are characterized by the ability to maintain self-sustaining ≈24 hr oscillations in constant conditions and to adjust the phase of those oscillations in response to environmental signals (e.g. transitions between light and dark). To represent the first feature, we model a circadian clock as an oscillator that runs along a 'light' limit cycle with speed $\omega_L$ when lights are on, and along a 'dark' limit cycle with speed $\omega_D$ when lights are off (***Appendix 1—figure 1A***). Because the oscillator is always on one of the limit cycles, its state can be fully described by a single variable, the phase $\theta(t)$:

$$\frac{d\theta}{dt} = \omega_L \quad \text{(in light)}$$

$$\frac{d\theta}{dt} = \omega_D \quad \text{(in dark)}$$

We define frequencies in units of cycles per hour (1 cycle = $2\pi$ radians), so that the duration of one cycle in constant conditions is $T = 1/\omega$ hours, and phase $\theta(t)$ is defined between 0 and 1.

We assume that the oscillator transitions between the two limit cycles immediately at dark-to-light ('dawn') and light-to-dark ('dusk') transitions, with a change in phase. We define $L(\theta)$ and $D(\theta)$ as phase-shifts accompanying clock responses to lights on and lights off cues, respectively:

$$\theta(t) \rightarrow \theta(t) + L(\theta(t)) \quad (\text{at dark} - \text{to} - \text{light})$$

$$\theta(t) \to \theta(t) + D(\theta(t)) \quad (\text{at light} - \text{to} - \text{dark}).$$

Though in principle $L(\theta)$ and $D(\theta)$ could be arbitrary periodic functions, our measurements in **Figure 3** suggest that $L(\theta)$ and $D(\theta)$ are approximately linear over the range of clock phases when dawn and dusk occurred in our experiments. Henceforth we consequently assume that $L$ and $D$ are linear functions of $\theta$:

$$L(\theta) = -l(\theta - \theta_L)$$

$$D(\theta) = -d(\theta - \theta_D).$$

Given this framework, we can now determine the change in clock state over any interval simply by adding the changes in phase at the transitions and accumulated during the free runs. We now explicitly compute the change in clock state in response to a dark pulse and the state of stable entrainment for light-dark cycles of arbitrary period.

## Modeling phase resetting

To probe the response of an oscillator, one can subject it to a perturbation of duration $\delta$ and compare the resulting phase to that of an unperturbed oscillator. Plotting the results of such an assay as a function of the phase $\theta_t$ at which the perturbation is delivered yields a phase response curve (PRC; **Appendix 1—figure 1C**, **Figure 5A**). Alternatively, varying the duration of the perturbation while delivering it at a fixed phase corresponds to a 'wedge' experiment (**Figure 5B**).

To determine the phase response, we need to determine the phase evolution as the system transitions to dark, free runs in the dark, and then transitions to light. Note that because this model treats the transitions between the light and dark cycles as instantaneous, we expect that it will only be valid for dark pulses that are long compared to the true relaxation time of the clock. Mathematically,

$$\theta'_{t+\delta,+} = \theta_t + D(\theta_t) + \delta\omega_D + L\left(\theta'_{t+\delta,-}\right),$$

where the prime denotes the perturbed oscillator, and the + and − in the subscripts distinguish the phase after and before the transition from dark-to-light at $t + \delta$. That is,

$$\theta'_{t+\delta,-} = \theta_t + D(\theta_t) + \delta\omega_D.$$

Substituting the linear forms above for $D$ and $L$ and grouping like terms,

$$\theta'_{t+\delta,+} = \theta_t[(1-d)(1-l)] + \delta(1-l)\omega_D + [(1-l)d\theta_D + l\theta_L].$$

Subtracting the unperturbed phase $\theta_{t+\delta} = \theta_t + \delta\omega_L$,

$$\Delta\theta = \theta'_{t+\delta,+} - \theta_{t+\delta} = \theta_t[ld - d - l] + \delta[(1-l)\omega_D - \omega_L] + [(1-l)d\theta_D + l\theta_L].$$

Thus we obtain a formula for the change in phase in response to a dark pulse, with linear dependences on the phase of the perturbation, $\theta_t$, and its duration, $\delta$.

## Modeling entrainment to light-dark cycles

We can similarly describe the phase of a clock stably entrained to light-dark cycles of varying day length. To this end, consider a phase oscillator subjected to light-dark cycles of period $T$ and day length $\tau$ (**Appendix 1—figure 1D**). In this case, rather than a phase difference, we need to determine $\theta_\tau$ such that $\theta_{T+\tau} = \theta_\tau + 1$. If this condition is met, the oscillator is stably entrained to a cycle of period $T$. In other words, we enforce periodicity.

Proceeding analogously to above, we determine the evolution over a full driving period, starting at dusk. This comprises a transition to the dark at time $\tau$, free run in the dark for duration $T - \tau$, a transition to the light at time $T$, and free run in the light for duration $\tau$. Mathematically,

$$\theta_{T+\tau} = \theta_\tau + D(\theta_\tau) + \omega_D(T - \tau) + L(\theta_T) + \omega_L\tau.$$

Again, substituting the linear forms above for $D$ and $L$ and grouping like terms,

$$\theta_{T+\tau} = \theta_\tau[(1-d)(1-l)] + T(1-l)\omega_D - \tau[(1-l)\omega_D - \omega_L] + [(1-l)d\theta_D + l\theta_L].$$

Setting $\theta_{T+\tau} = \theta_\tau + 1$ and solving for $\theta_\tau$,

$$\theta_\tau = \tau\frac{(1-l)\omega_D - \omega_L}{ld - d - l} - T\frac{(1-l)\omega_D}{ld - d - l} - \frac{(1-l)d\theta_D + l\theta_L - 1}{ld - d - l}.$$

Again, we obtain a formula with linear dependence, here on the day length $\tau$ and the driving period $T$.

## Comparison with experiment

The expressions above describe phases, but we do not measure phase directly. Rather, we measure the time of peak expression of a reporter. Thus it is necessary to convert from phase to the time of peak expression. To this end, we arbitrarily assign the peak expression to occur at $\theta = 0.5$ (midway through the cycle). Because phase advance and delay correspond to earlier and later peak times respectively, the time of the peak shifts with opposite sign to the phase. In each case above, we are interested when the peak (or $\theta = 0.5$) occurs relative to the last dawn (i.e. immediately after the action of $L$). We can solve for this peak time:

$$t_{pk} = \frac{0.5 - \theta_x}{\omega_L},$$

where $x = t + \delta$ (end of dark pulse) in the case of phase resetting and $x = 0$ (last dawn) in the case of entrainment. To solve for $t_{pk}$ in entrained conditions, we make use of the fact that $\theta_0 = \theta_\tau - \omega_L\tau$, allowing us to substitute the expression above for $\theta_\tau$. For phase resetting, we must apply this formula to both the perturbed and unperturbed phase and then subtract:

$$\Delta t_{pk} = t_{pk}\left(\theta'_{t+\delta+}\right) - t_{pk}(\theta_{t+\delta}) = -\Delta\theta/\omega_L.$$

Furthermore, we note that certain combinations of the frequencies $\omega_L$ and $\omega_D$ and the linear sensitivities $l$ and $d$ occur repeatedly. We thus define

$$\beta_1 = \frac{\omega_D}{\omega_L}\left[\frac{(1-l)}{ld - d - l}\right]$$

$$\beta_2 = -\frac{1}{ld - d - l}.$$

Then, for phase resetting

$$\Delta t_{pk} = \theta_t\left(\frac{1}{\omega_L \beta_2}\right) + \delta\left(1 + \frac{\beta_1}{\beta_2}\right) + C_1$$

and for entrainment

$$t_{pk} = \tau(1 - \beta_1 - \beta_2) + T\beta_1 + C_2,$$

where $C_1$ and $C_2$ are constants that do not depend on $\theta_t$, $\delta$, $\tau$, or $T$.

In the global fits for all datasets in **Figure 5A–C**, we varied $\beta_1$ and $\beta_2$ to simultaneously fit $t_{pk}$ to our phase-resetting and wedge data and $t_{pk}$ to our seasonal entrainment data. Thus, the slopes of all the linear fits in **Figure 5A–C** were derived from the same values of $\beta_1$ and $\beta_2$. See Computational Methods for further details.

## Geometric resetting

In our experimental measurements, we observed $L$ and $D$ step-response functions with linear regions with slopes < 1. Here we present a simple dynamical systems picture where a tunable, linear dependence of phase shift magnitude on oscillator phase arises from the geometry of how a strongly attracting limit cycle is deformed by a changing external input.

We suppose that the oscillator is described by a circular limit cycle in the plane with unit radius during the day. During the night, we assume that the limit cycle remains in the plane, but is offset a distance $X$ from the daytime cycle and has an altered radius $R$ (**Figure 6A**). At the moment of the light-to-dark transition, the state of the system is on the daytime limit cycle—it is then attracted to the nighttime limit cycle. The step-response function $D(\theta)$ describes the shift between the new phase angle (measured relative to the center of the nighttime limit cycle) and the original phase $\theta$ (**Figure 6A**).

If we place the origin at the center of the nighttime limit cycle, the Cartesian coordinates of the system state immediately before the light-to-dark transition, when the system is still on the daytime orbit, are:

$$(\cos\theta + X, \sin\theta)$$

If we assume that the fixed point giving rise to the nighttime orbit is very strongly attracting, then the system approaches the limit cycle much faster than any circulation occurs. In this limit, the light-to-dark transition leads to a purely radial jump from the daytime orbit towards the center of the nighttime orbit (**Figure 6A–B**). Under these conditions, the new phase $\theta_{night}$ on the nighttime cycle is simply the angle measured relative to the center of the new limit cycle:

$$\theta_{\text{night}} = \tan^{-1}\left(\frac{\sin\theta}{\cos\theta + X}\right)$$

To develop intuition about how this mapping between phase on day and night orbits is affected by their geometric arrangement, we can perform a Taylor expansion for angles near $\theta = 0$ at dusk, corresponding to conditions close to LD 12:12:

$$\theta_{\text{night}} \approx \frac{\theta}{1+X}$$

Likewise, it can be shown that for transitions from nighttime to daytime cycles, near $\theta = \pi$, the phase immediately after transition to the daytime cycle at dawn can be given by:

$$\theta_{\text{day}} = \pi + \frac{\theta - \pi}{1 + X/R}$$

These expressions show that, in the linear portion of the step response function, angles are compressed by a factor tunable by the relative geometry of the light and dark limit cycles. In this way, the geometric arrangement of the limit cycles sets the slopes of $L(\theta)$ and $D(\theta)$. Numerical simulations in *Figure 6* and *Figure 6—figure supplement 1* were carried out using trigonometric formulas as above to convert phases between the daytime and nighttime limit cycles. Details of simulations in *Figure 6—figure supplement 2* are described in Computational Methods.

