## [Decision Letter]

Thank you for submitting your article "The cyanobacterial circadian clock follows midday in vivo and in vitro" for consideration by *eLife*. Your article has been reviewed by two peer reviewers, and the evaluation has been overseen by Naama Barkai as the Senior Editor and Reviewing Editor. The reviewers have opted to remain anonymous.

The reviewers have discussed the reviews with one another and the Reviewing Editor has drafted this decision to help you prepare a revised submission.

Summary:

The authors examine how the cyanobacterial clock aligns under varying environmental cycles, revealing a simple scaling law to midday across various Light/Dark cycles. In a very nice follow up they then show that it is possible to recreate these results in vitro using metabolic pulses. It is an impressive strength of the paper that they are able to go from in vivo to in vitro and obtain similar results. They go on to develop a mathematical model, based on the immediate response of the clock to environmental changes, which they then successfully test. The model is elegant and simple and produces experimentally testable predictions that the authors follow up on.

Essential revisions:

1) Please downplay the optimality argument, as requested by the reviewer below. Both reviewers agreed that this argument is not well supported.

2) Most importantly, the experiments lack biological repeats. Please either justify why biological repeats are not needed, or (largely recommended) repeat the experiments to verify reproduciblity. This is important in particular for the in vitro experiments (Figure 2 and 3), where it is stated that the experiment was done once (subsection “Estimation of clock phase in vitro in metabolic cycles”, last paragraph and subsection “Non-parametric bootstrapping of step-response datasets and *L* and *D* functions”, first paragraph), but also for the other 96-well based experiments.

*Reviewer #1:*

The authors study the phase of the cyanobacterial circadian oscillator under entrainment with different photoperiods (day length). A main finding is that the internal clock phase tracks midday under entrainment, both in vivo and in the test tube, and this can be explained by the measured phase shifts at dawn and dusk. Linearity properties of the phase shifting curves in a physiological range are sufficient to explain midday tracking, but this also generalizes to more complex perturbations. Overall, an elegant and general picture emerges, and a simple geometrical interpretation is proposed: that of the deformation of the clock orbit in light vs. dark conditions.

Would there be a more powerful way to exploit the purF reporter? I assume the authors do not have dual reporter strains? I would recommend putting the purF figure in the main, and to analyze the KaiC data in the same way (i.e. multiple peaks etc.).

An important test for the model would be to analyze period mutants. That, is assuming that the phase shifts do not depend on the mutations, it would be straightforward to predict if mode-locking can still occur, and if so what the entrainment phase is. Looking at period mutants seems quite obvious, why did the authors not do this?

Moreover, it would have been informative to identify mutants with entrainment defects. Or perhaps some entrainment mutants are already reported, and could have been analyzed?

Also, from the dynamical systems points of view, 1-d maps like the one proposed exhibit complex behavior (period doubling, chaos, etc.). It would have been very interesting to try and probe whether other regimes besides 1:1 can occur.

*Reviewer #2:*

The work appears to be of high quality, has very robust data, and should be interesting to the *eLife* audience. I do have the following concerns however.

1) Concepts developed in the paper. In the paper it is assumed that clocks in individual cells can only track one phase, and it is argued that tracking midday should be the optimal strategy, as it allows proportional expression of dawn and dusk genes. I think both these ideas require further thought and justification. It is well known in the plant clock field that the clock network can track dawn and dusk. For example (Edwards et al., 2010). There has also been theoretical studies on clocks (without assuming any multicellularity or coupling between cells) to examine what it takes to track more than one phase (Rand et al., J. Royal Society Interface, 2004). As the cyanobacterial clock is limited to one feedback loop this presumably means it is limited to tracking one phase, but this point should be discussed. If you are going to track one phase, it is not clear that tracking midday is an optimal decision. It could be that evening genes become more important under shorter days for example. It is not clear to me that equi-partitioning of the resources into two classes is necessarily clearly the optimal strategy.

2) Figure 7F: The cycle in phosphorylation state for the 'Day cycle' in the invitro system is quite elliptical in shape. It is not clear how one would estimate *X* and *R* from this ellipse, and whether or not one would gain a linear phase shift relationship in L(θ) from the elliptical orbits. It would be useful if the authors could comment on this, and perhaps show how the experimental orbits would map to the calculated D(θ) and L(θ) from the geometrical model.

[Editors' note: further revisions were requested prior to acceptance, as described below.]

Thank you for resubmitting your work entitled "The cyanobacterial circadian clock follows midday in vivo and in vitro" for further consideration at *eLife*. Your revised article has been favorably evaluated by Naama Barkai as the Senior Editor and Reviewing Editor, and two reviewers.

Please see comments below – the reviewers would like you to emphasize aspects related to the reproducibility, as was raised in the previous review. Specifically, please show all that data in the main figures (in Figure 2C, there's plenty of space to plot the two additional in vitro measurements). Also, please adapt your interpretation to take into the observed variability and discuss what the potential sources are.

*Reviewer #1:*

I believe the revised version addresses the general editorial comments adequately.

In addition, though the authors have partially addressed my comments, I regret that they did not consider my first major point on integrating the purF data in the main text, and also not analyze their KaiC data in the same way. I liked the purF analysis with the multiple peaks, which seemed more thorough than that presented for KaiC in the main. In particular this modification seemed not to represent a lot of extra work, so it is not intuitive why it was not done.

*Reviewer #2:*

In general I am happy with the revision, but I have the following concerns.

The new in vitro experiments slopes using a different method based on a fluorescent probe are less near to m = 0.5 than one would have expected (at 0.38). It isn't clear why the non-repeated in vitro study based on% KAICP is used in Figure 2 (with a m value of 0.51), whilst the new method that is repeated (showing an answer that is less close to the in vivo values) is in the supplement. This is especially true as Figure 3 now uses the new data from the fluorescent probe. In Figure 2C the comparison is only made between the non-repeated in vitro method based on %P KaiC, but the slopes based on the new method using an in situ fluorescent probe are not included. The authors should include the values from the new experiments in Figure 2C.

In Figure 3A it appears that phase shifts much larger than 4 hours are possible, even though phase shifts of <4 hours are claimed in the text and displayed in Figure 3C. Could the authors please explain the differences?

---

## [Author Response]

*Essential revisions:*

*1) Please downplay the optimality argument, as requested by the reviewer below. Both reviewers agreed that this argument is not well supported.*

We agree with the reviewers that the argument that biosynthetic resources need to be equally partitioned by the clock between morning and afternoon gene expression programs is not strongly supported by our data. Thus we have shortened the discussion of this hypothesis and moved it to the Discussion section of the paper. Likewise, we have moved the figure panels explaining the idea to Figure 7—figure supplement 3.

*2) Most importantly, the experiments lack biological repeats. Please either justify why biological repeats are not needed, or (largely recommended) repeat the experiments to verify reproduciblity. This is important in particular for the invitro experiments (Figure 2 and 3), where it is stated that the experiment was done once (subsection “Estimation of clock phase in vitro in metabolic cycles”, last paragraph and subsection “Non-parametric bootstrapping of step-response datasets and L and D functions”, first paragraph), but also for the other 96-well based experiments.*

We have included additional, independent in vitro measurements in the paper and clarified the number of biological replicates for the in vivo results (3 independent measurements of entrainment scaling, as detailed in new Table 1).

The in vitro entrainment experiment in Figure 2 has been repeated two more times with a new batch of purified proteins and measured using a fluorescent reporter of oscillations which allows us to more accurately determine phase. These data, along with validation of the fluorescent reporter system, are now in Figure 2—figure supplements 1 and 2.

We measured the L(θ) and D(θ) step response functions in two more replicates, again using a new set of purified proteins and the fluorescent reporter system. These data are now shown in Figure 3. The original data from Figure 3 (KaiC phosphorylation measured by SDS-PAGE) is now in Figure 4—figure supplement 2. The modeling in Figures 4,6 has also been updated to use these more accurately determined L(θ) and D(θ) step response functions for the new protein prep. While replicate measurements on the same protein prep are in close agreement (curves in Figure 3C and 3D) there is some prep-to-prep variability in these experiments; our new protein prep is less responsive to nucleotide shifts. For both preparations of clock proteins, the model works well in the sense that phase shift functions correctly predict linear scaling of entrainment (Figure 4C).

*Reviewer #1:*

*[…] Would there be a more powerful way to exploit the purF reporter? I assume the authors do not have dual reporter strains? I would recommend putting the purF figure in the main, and to analyze the KaiC data in the same way (i.e. multiple peaks etc.).*

We decided to keep the purF reporter data in the supplement as in the original submission. As the reviewer suggests, the best comparison would use a dual reporter approach, which is not currently feasible using our bioluminescence detection system.

*An important test for the model would be to analyze period mutants. That, is assuming that the phase shifts do not depend on the mutations, it would be straightforward to predict if mode-locking can still occur, and if so what the entrainment phase is. Looking at period mutants seems quite obvious, why did the authors not do this?*

We agree that varying the natural frequency of the clock vs. the driving cycle is an important test for the model. Our preliminary experiments with period mutants suggested that many known point mutants that alter the period of oscillations also alter their amplitude. In light of our geometric model, where amplitude changes correspond to altered size of the limit cycle attractor, we expect that mutants that change both period and amplitude will change the entrained behavior of the clock in a way which requires a case-by-case investigation of each mutant. For this reason, we probed this issue in the reverse way, by changing the frequency of the driving cycle for WT cells. These data are in Figure 6C along with a simultaneous fit showing self-consistency of the model (unchanged from the original submission).

*Moreover, it would have been informative to identify mutants with entrainment defects. Or perhaps some entrainment mutants are already reported, and could have been analyzed?*

This is a good suggestion, though most known entrainment mutants (e.g. cikA, glgC, pr-1 allele of kaiC) seem to act in part by altering metabolism in light vs. dark, and will again need to be studied in a case-by-case way to understand how these effects can be translated into the dynamical systems language of attractor geometry. We think it’s a good direction for future work.

*Also, from the dynamical systems points of view, 1-d maps like the one proposed exhibit complex behavior (period doubling, chaos, etc.). It would have been very interesting to try and probe whether other regimes besides 1:1 can occur.*

We have now generated a bifurcation diagram as a function of driving period for the model based on the *L* and *D* maps derived from our biochemical measurements. This map is in Figure 3—figure supplement 1. As expected, we find a band of frequencies around the natural frequency where entrainment occurs (an Arnold tongue). Outside of this band, apparently chaos-like dynamics emerge, though entrainment reappears at multiples of the natural frequency (e.g. near 12 hour driving period).

*Reviewer #2:*

*The work appears to be of high quality, has very robust data, and should be interesting to the eLife audience. I do have the following concerns however.*

*1) Concepts developed in the paper. In the paper it is assumed that clocks in individual cells can only track one phase, and it is argued that tracking midday should be the optimal strategy, as it allows proportional expression of dawn and dusk genes. I think both these ideas require further thought and justification. It is well known in the plant clock field that the clock network can track dawn and dusk. For example (Edwards et al., 2010). There has also been theoretical studies on clocks (without assuming any multicellularity or coupling between cells) to examine what it takes to track more than one phase (Rand et al., J. Royal Society Interface, 2004). As the cyanobacterial clock is limited to one feedback loop this presumably means it is limited to tracking one phase, but this point should be discussed. If you are going to track one phase, it is not clear that tracking midday is an optimal decision. It could be that evening genes become more important under shorter days for example. It is not clear to me that equi-partitioning of the resources into two classes is necessarily clearly the optimal strategy.*

We agree with the reviewer that it is not obvious that an organism with a single feedback loop clock should track the middle of the day. We have removed our speculative hypothesis about metabolic balance from the beginning of the paper and instead raise the idea briefly in the Discussion section. We have now included citations to this previous work in the following paragraph:

*“*The ability of circadian systems to keep track of the phase of the light-dark cycle has been long recognized in plants, insects, rodents and higher mammals (de Montaigu et al., 2015; Daan et al., 2001; Edwards et al., 2010; Hut, Oort and Daan, 1999; Hut et al., 2013; Wehr 2001). […] Consistent with this picture, we find that the core circadian oscillator in cyanobacteria, which relies on a single posttranslational feedback loop, keeps track of a single phase—the midpoint of the day portion of the cycle….”.

*2) Figure 7F: The cycle in phosphorylation state for the 'Day cycle' in the invitro system is quite elliptical in shape. It is not clear how one would estimate X and R from this ellipse, and whether or not one would gain a linear phase shift relationship in L(θ) from the elliptical orbits. It would be useful if the authors could comment on this, and perhaps show how the experimental orbits would map to the calculated D(θ) and L(θ) from the geometrical model.*

We agree that the model in Figure 7 with the assumption of circular, strongly attracting limit cycles is unrealistically simple. We have added new material in Figure 7—figure supplement 2 that shows the results of relaxing these assumptions. We consider a finite relaxation time when the system is displaced from the limit cycle following a dusk or dawn transition, ellipticity, and nonuniform angular velocity. We find that the entrainment behavior is very similar until the relaxation time becomes comparable to the length of the day itself. Similarly, ellipticity or nonuniform rotational velocity have minor effects, but do not change the basic conclusion that limit cycle separation in day vs. night must be comparable to the radius of the larger orbit to obtain entrainment similar to our observations in *S. elongatus*.

[Editors' note: further revisions were requested prior to acceptance, as described below.]

*Please see comments below – the reviewers would like you to emphasize aspects related to the reproducibility, as was raised in the previous review. Specifically, please show all that data in the main figures (in Figure 2C, there's plenty of space to plot the two additional in vitro measurements). Also, please adapt your interpretation to take into the observed variability and discuss what the potential sources are.*

We have moved all of our in vitro measurements of entrainment and step-response functions into the main panels in Figures 2 and 3. We’ve added the following text to the Discussion interpreting our results in light of the variability we observe:

“The step-response curves underlying entrainment in our model are nonlinear functions of clock phase, but they can be successfully approximated by linear functions over the interval of clock phases used during entrainment. […] In this study, we achieve this by measuring free-running oscillations in an automated way using the fluorescence polarization probe.”

Additionally, we have modified the text in the Results section to discuss variability in our in vitro measurements and its possible sources:

“As mentioned above, we also measured L(θ) and D(θ) step-response functions with a separate preparation of Kai proteins using a gel-based assay to read out the phase of the KaiC phosphorylation rhythm (Figure 3E-F). […] The discrepancy in magnitude between these measurements may point to differences in sensitivity to input cues between different preparations of Kai proteins, or to a slight perturbative effect of fluorescently labeled KaiB.”

Taking into account our observed variability, we have also modified the text to deemphasize the specific value of the scaling coefficient, *m*; that *m* takes intermediate values both in vivo and in vitro shows that the reconstitution captures the essential physics of the system, but we have no reason to expect the in vivo and in vitro phase responses to be precisely the same:

“To more accurately measure the scaling of entrained phase with day length in KaiABC oscillator reactions, we turned to a fluorescence polarization probe that enables automated measurement of oscillator state with high temporal resolution (Figure 2— figure supplement 1) (Chang et al., 2012; Heisler et al., in press). […] Despite the variability in our estimates of *m*, these measurements argue that the in vitro oscillator successfully captures the essential feature of seasonal entrainment we observed in vivo: linear scaling of entrained phase with an intermediate slope.”

We have added text explaining why we have higher confidence in our fluorescence polarization measurements than in the traditional SDS-PAGE analysis of KaiC phosphorylation:

“We found that simulated entrainment of the phase oscillator model was particularly sensitive to the regions surrounding subjective dusk (on D(θ)) and subjective dawn (on L(θ)) (Figure 3E-F, *colored arrows*), which were only sampled with small numbers of points. Because the fluorescence polarization approach allows us to measure many conditions in an automated way over many days, and thus to disentangle phase shifts from period differences (Figure 3C-D), these higher time resolution measurements better constrain the portions of the response functions critical for entrainment.”

*Reviewer #1:*

*I believe the revised version addresses the general editorial comments adequately.*

*In addition, though the authors have partially addressed my comments, I regret that they did not consider my first major point on integrating the purF data in the main text, and also not analyze their KaiC data in the same way. I liked the purF analysis with the multiple peaks, which seemed more thorough than that presented for KaiC in the main. In particular this modification seemed not to represent a lot of extra work, so it is not intuitive why it was not done.*

We agree with the reviewer that locally fitting parabolas to mark peak positions is a good way to estimate peak times of our oscillations. We have done this analysis for the strain carrying the kaiBC reporter and have replaced data in Figure 1C with peak times obtained using this approach.

In particular, we found good agreement in the value of the scaling coefficient m calculated using the sinusoidal fitting (m=0.55 ± 0.02) and parabolic fitting approaches (m=0.53 ± 0.01). Our estimates of the slope m for the kaiBC expression reporter are listed in Table 1.

We feel that adding the purF data to Figure 1C would distract from Figure 1’s focus on the expression of the kaiBC gene reporter and, in turn, the main narrative. Interested readers can find these data in Figure 1—figure supplement 3.

*Reviewer #2:*

*In general I am happy with the revision, but I have the following concerns.*

*The new in vitro experiments slopes using a different method based on a fluorescent probe are less near to m = 0.5 than one would have expected (at 0.38). It isn't clear why the non-repeated in vitro study based on% KAICP is used in Figure 2 (with a m value of 0.51), whilst the new method that is repeated (showing an answer that is less close to the in vivo values) is in the supplement. This is especially true as Figure 3 now uses the new data from the fluorescent probe. In Figure 2C the comparison is only made between the non-repeated in vitro method based on% KAICP, but the slopes based on the new method using an in situ fluorescent probe are not included. The authors should include the values from the new experiments in Figure 2C.*

We agree with the reviewer’s suggestion and have added all of the data into main figure panels (see above).

*In Figure 3A it appears that phase shifts much larger than 4 hours are possible, even though phase shifts of <4 hours are claimed in the text and displayed in Figure 3C. Could the authors please explain the differences?*

The reviewer raises an excellent point that there is a visual discrepancy between the phase shift magnitudes in Figure 3A and 3C. The reason for this difference is that there is a ≈2 hour difference in oscillator periods in day-mimicking versus night-mimicking conditions. Thus, differences in peak times in Figure 3A, which shows oscillations over a 72 hour time period after the step perturbation, have two contributions: (1) the instantaneous phase shift caused by the step and (2) the two rhythms drifting out of phase over time as a result of the different periods of the control and step-down reactions. Thus the peak-to-peak time differences in Figure 3A are larger than in Figure 3C, which displays our estimates of instantaneous phase shifts at the time of the step and excludes any contributions from period differences accumulated in free-running conditions after the step transition.

In addressing the reviewer’s comment, we refined our procedure for computing phase shifts, as described in the third paragraph of the subsection “Seasonal adaptation of the circadian oscillator can be decomposed into step responses to individual metabolic cues”. While this shifts certain numbers slightly, it does not have any significant impact on the results or their interpretation. To avoid confusion, we have clarified how these measurements are described in the main text and have added a new supplementary figure (Figure 3—figure supplement 1) illustrating the data analysis procedure:

“We proceeded to calculate phase shifts from these measurements, taking into consideration that a switch to nighttime buffer conditions could affect the oscillator in two ways: the phase of the oscillation can shift, and the oscillator period may change, both of which can change the relative timing of peaks. [...] We detail our procedure for analyzing these data to extract phase shifts in Figure 3—figure supplement 1.”